# Major waves of H2A.Z incorporation during mouse oogenesis and preimplantation embryo development

Madeleine Fosslie [1,2,10], Erkut Ilaslan [2,3,10], Trine Skuland [4,5], Michel Choudalakis [1,2], Mirra Søegaard[1,2,3], Jason Alexander Halliwell [3], Shaista Khan[6], Marie Indahl[1,2,4], Maria Vera-Rodriguez [2,4], Adeel Manaf [1,2], Annika Vera Geijer-Simpson[7], Rajikala Suganthan[1,2], Ingunn Jermstad[6], Knut Tomas Dalen [6], Ragnhild Eskeland [8,9], Elisabeth Kommisrud [7], Arne Klungland [1,2], Magnar Bjørås [1,2], Peter Fedorcsak[2,4,5], Gareth D. Greggains[1,2,4], Mika Zagrobelny [2,3] ✉, John Arne Dahl [1,2] ✉ & Mads Lerdrup [2,3] ✉

Epigenomes of mammalian oocytes and embryos undergo major transitions essential for successful development. Here, we provide genome-wide maps of histone variant H2A.Z during twelve stages of mouse oogenesis and preimplantation embryo development and relate it to histone marks and genomic features. This revealed that major waves of H2A.Z incorporation occur early in growing oocytes, forming distinct patterns of maternal, embryonic, and persistent H2A.Z enrichment. Late maternal enrichment is inherited by the zygote and precedes reduced formation of lamina associated domains and early replication in the maternal genome of 2-cell embryos. Persistent H2A.Z enrichment is strongly associated with CpG islands and H3K4me3 near transcription start sites of active genes, but thousands of maternal and embryonic H2A.Z incorporation sites exist elsewhere, frequently at transposable elements. The persisting H2A.Z enrichments across related developmental stages enable preservation of epigenetic information despite major concurrent changes in H3K4me3, H3K27me3, and DNA methylation. Altogether, this advances our understanding of how histone variants contribute to epigenetic reprogramming during mammalian oogenesis and early development.

Mammalian oocytes and preimplantation embryos undergo extensive epigenetic changes, involving the establishment and remodeling of specific epigenetic marks crucial for development and fertility, reviewed in[1]. The histone variant H2A.Z is essential for embryo development, as *H2A.Z* knockout (KO) mouse embryos fail to develop beyond embryonic day 6.5[2]. H2A.Z has approximately 60% amino acid similarity to core histone H2A and is deposited in a cell-cycle independent manner, reviewed in[3]. H2A.Z can alter the structure and stability of chromatin and nucleosomes, recruits different types of reader proteins, and provides an obstacle for transcriptional elongation of the RNA pol II complex, reviewed in[4]. Accordingly, H2A.Z is involved in diverse biological processes, such as gene activation, firing of replication origins, DNA repair, meiotic recombination, embryonic stem cell differentiation, nucleosome turn-over, memory formation, chromosome segregation, heterochromatin silencing, and progression through the cell cycle[3,5,6]. H2A.Z exists as two isoforms in mice (H2A.Z.1 and H2A.Z.2), which are products of two nonallelic genes (*H2afz* and *H2afv*) that encode proteins differing by only three amino acids and

are differentially expressed across tissues[7–9]. In addition, the function of H2A.Z is affected by post-translational modifications and nucleosome partners[3].

H2A.Z is generally present at promoters with CpG islands (CGIs) and at transcription start sites (TSSs) of active and poised genes in various cell types, including stem cells[10]. In mammals, it usually colocalizes with the histone modifications H3K4me3 and H3K27me3[11–13]. H2A.Z can affect the deposition of H3K4me3 and H3K27me3, as it can facilitate recruitment of MLL and PRC2 complex proteins, which are responsible for the formation of H3K4me3 and H3K27me3, respectively[14]. The H2A.Z protein was shown to be present in MII oocytes[9], but a recent genome-wide study did not detect genomic H2A.Z enrichment by ULI-NChIP-seq in MII oocytes despite a strong immunostaining signal[15]. Additionally, knockdown of *H2A.Z* at the zygote stage significantly reduces blastocyst formation, but has no effect on morula development, leading to the hypothesis that H2A.Z is not essential for initiation of zygotic genome activation (ZGA) in mice[15]. However, since the genome-wide dynamics of H2A.Z have not yet been analyzed in detail in developing mammalian oocytes, much remains to be learned about the possible roles at these stages.

Concurrent with the epigenetic changes in developing oocytes and embryos, transposable elements (TEs) are expressed in stage-specific waves, indicating precise regulation[16]. TEs are mobile multi-copy DNA sequences that account for approximately half of the mammalian genome and are generally silenced to prevent indiscriminate activation and mutations by transposition, reviewed in[17]. Nonetheless, TEs are thought to promote evolution and facilitate organismal development by contributing enhancer and promoter sequences, modifying three-dimensional (3D) chromatin architecture, and giving rise to novel regulatory genes[17]. Among TEs, retrotransposons (RTs) constitute up to 38% of the transcriptome in mammalian preimplantation embryos[18]; indeed, early development is marked by expression of the RTs ERVL, MaLR, and LINE1, some of which are fundamental for the progression of mammalian embryonic development, reviewed in[19,20].

During mouse oogenesis, primordial follicles develop shortly after birth and mature into primary follicles. Upon activation, the oocyte within each follicle grows and increases its transcription during the subsequent postnatal (P) period, reviewed in[21]. The growing follicles reach the antral stage when a fluid-filled cavity forms (Fig. 1a). Inside the oocyte nucleus, known as the germinal vesicle (GV), chromatin condenses around the nucleolus. This transition marks a shift in chromatin organization from a non-surrounded nucleolus (NSN) to a surrounded nucleolus (SN) configuration. Concurrently, global transcription is silenced, and the oocyte reaches its full size around this time. In preovulatory follicles, oocytes resume meiosis in response to hormonal signals and progress to metaphase II (MII), where they are considered mature and remain arrested until ovulation and fertilization occur[21]. Nuclear organization is established de novo shortly after fertilization, reflected in the formation of Lamina Associated Domains (LADs) where the genome interacts with the nuclear lamina[22]. The fertilized oocyte relies on maternal RNAs until transcription is initiated during minor and major zygotic genome activation (ZGA) at the zygote and 2-cell stages, respectively, reviewed in[23].

Given the relative scarcity of mouse oocytes and the limited sensitivity of chromatin immunoprecipitation and sequencing (ChIP-seq) assays, the genomic localization of H2A.Z and its relationship to the epigenome and transcriptional regulation in oocytes has remained unstudied. Here, we carry out picogram-scale chromatin immunoprecipitation and sequencing (picoChIP-seq) to provide genome-wide maps of H2A.Z throughout mouse oogenesis by analyzing six oocyte stages, spanning from just after the oocytes have entered the growth phase until mature MII oocytes. Additionally, we analyze six preimplantation embryo stages from zygote to blastocyst. In contrast to the distinctive redistribution of many histone marks[24], H2A.Z retains many canonical features during early development, including narrow peaks at TSSs and CGIs. Additionally, our maps reveal dynamic and stage-specific changes in H2A.Z deposition during oogenesis and early embryo development. Much of the stage-specific H2A.Z incorporation occurs at non-TSS and non-CGI loci often found at specific types of TE, and we find that late oocyte H2A.Z enrichment is inherited by the zygote. Interestingly, maternal H2A.Z is incorporated into regions in the genome where the embryonic formation of LADs differs markedly between the maternal and paternal genomes. Moreover, these loci replicate earlier in the maternal than the paternal genome, indicating that maternal H2A.Z incorporation or associated features may be instructive for the replication timing in the embryo[25]. Finally, we relate H2A.Z incorporation to genetic and epigenetic features and find an intricate relationship between H2A.Z, CpG density and H3K4me3. By characterizing these dynamic changes in H2A.Z distribution, this work highlights how histone variants contribute to the dynamic epigenetic landscape and genome regulation during oogenesis and preimplantation embryo development.

## Results

### H2A.Z incorporation in growing oocytes and early embryos

To assess the genome-wide profile of H2A.Z in limited numbers of mouse oocytes and early embryos, we utilized our recently developed picoChIP-seq method to target H2A.Z (Manaf et al., manuscript in revision). We further validated the efficiency of picoChIP-seq for H2A.Z by successfully mapping H2A.Z in mouse ES cell chromatin from 1000 and 500 cells, showing a comparable signal to that of published bulk H2A.Z ChIP-seq in terms of FPKM (Fragments per Kilobase pair per Million Reads) normalized and Z-transformed signal (relative to the entire genome)[14,26] (Supplementary Fig. 1a–c). Next, we mapped H2A.Z in oocytes from mice at postnatal day 7 (P7), P10 and P12, and in NSN, SN, and MII oocytes, as well as in zygotes (~4 h post-fertilization), 2-cell (~24 h post-fertilization), 4-cell, 8-cell, morula and blastocyst preimplantation embryos (Fig. 1a, Supplementary Data 1). This enabled us to resolve the oocyte GV stage and investigate whether the transition from the actively transcribing NSN-stage to the silenced SN-stage[21] causes any changes in H2A.Z localization. As H2A.Z ULI-NChIP-seq enrichment profiles have been generated for mouse MII oocytes and early embryos, we first carried out cross-validation of our H2A.Z picoChIP-seq data with this published data[15] (Supplementary Fig. 1d–f). As reported by the authors, the H2A.Z signal was virtually absent until ZGA, in contrast to our data. However, our H2A.Z signal correlated well with that from the 8-cell stage and onwards, both when assessing the signal at TSSs (Spearman's $\rho > 0.87$) and throughout the entire genome ($\rho > 0.73$ $\rho$, Supplementary Fig. 1d, e). During the revision process of this work, two additional studies investigating the role of H2A.Z in mouse oocytes were published[27,28], and we therefore performed further comparisons to the data generated in these. Considering the methodological differences between crosslinked and native ChIP[29] and the relatively low correlation with earlier data from MII and zygotes[15], we observed high genome-wide correlation in H2A.Z signal between our data and that of native ChIP-seq from related stages presented in these studies (Supplementary Fig. 2a, b). Additionally, we clustered and compared TSS-associated H2A.Z signal from several shared stages between the three published studies and our data (Supplementary Fig. 2c). This revealed TSS H2A.Z profiles that were largely identical in our study and the published works. However, our crosslinked ChIP-seq generally resulted in less of a gap directly at TSSs and provided stronger enrichment for P10 to 8-cell stages.

To investigate the location of H2A.Z in the mouse mm10 genome, we visualized the signal at gene bodies and unique transcription start sites (TSSs) (Fig. 1b). H2A.Z was clearly enriched at TSSs (Fig. 1b) and varied during different developmental stages (Fig. 1b, c). Of note, we observed a major wave of H2A.Z establishment in growing oocytes, with minimal presence in P7, rapidly increasing to peaking levels in P12

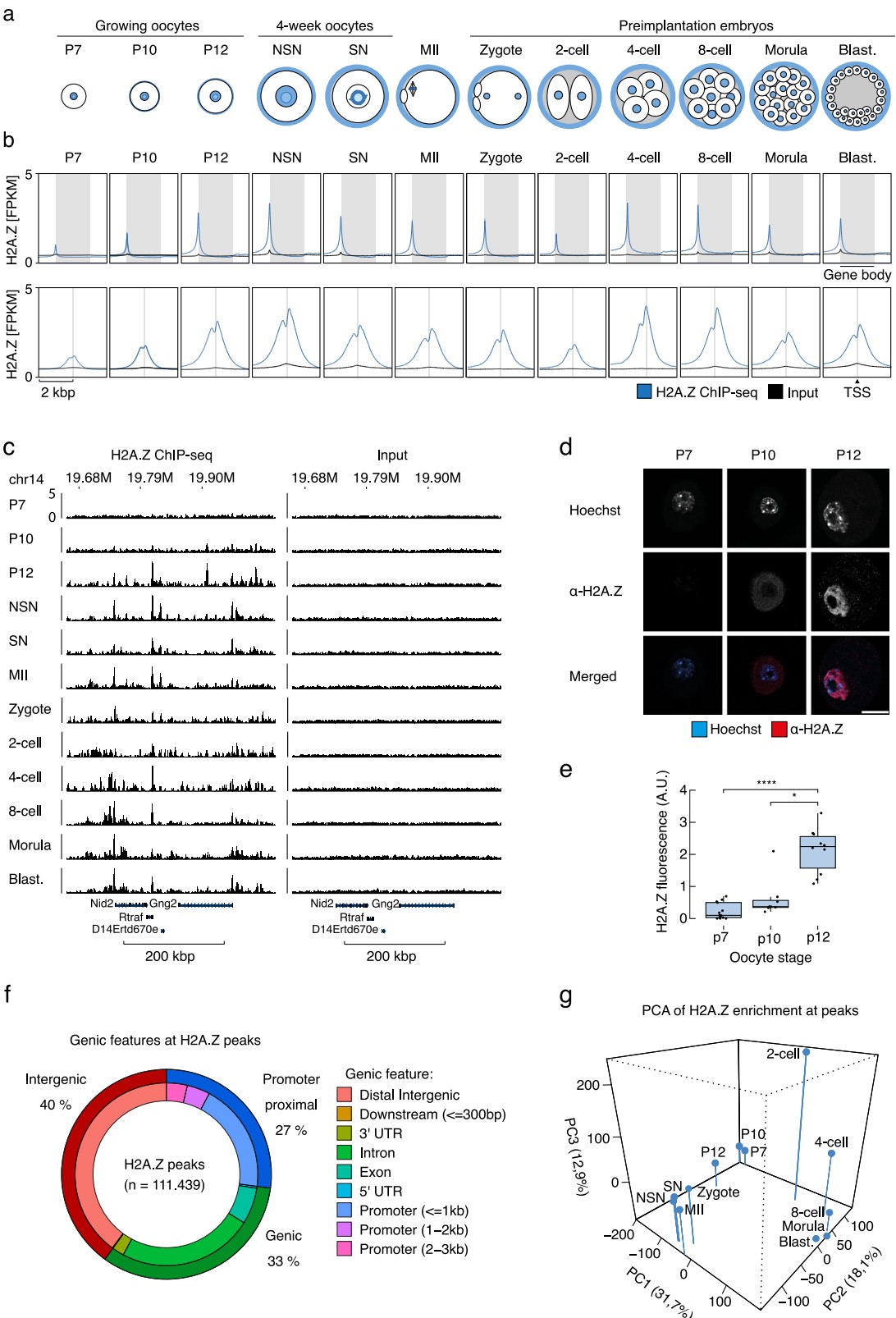

and NSN oocytes (Fig. 1b). This reveals a profound switch in the incorporation of a histone variant in mammalian oocytes. In addition, we observed a general transient reduction in relative H2A.Z enrichment at TSSs in 2-cell embryos (Fig. 1b). To validate the gain in H2A.Z during oocyte maturation through the P7 to P12 stages, we performed immunofluorescence, confocal microscopy, and quantification of the fluorescence intensities (Fig. 1d, e). This revealed a strong and highly

significant increase in H2A.Z fluorescence from P7 to P12 oocytes (Fig. 1e), reflecting the increases in H2A.Z enrichment observed from the ChIP-seq data.

To identify regions in the genome with high H2A.Z enrichment, we called peaks for each oocyte and embryo stage and generated a combined peak set (n = 111,439, Supplementary Data 2) that included all peaks from all developmental stages, except for the zygote to 4-cell

**Fig. 1 | H2A.Z is incorporated both in mouse oocytes and preimplantation embryos. a** Overview of the developmental stages that were sampled and analyzed in this study. P, postnatal day; NSN, non-surrounded nucleolus; SN, surrounded nucleolus; MII, metaphase II; Blast., blastocyst. **b** H2A.Z ChIP-seq signal in different developmental stages at gene bodies (shaded area, upper panel), $n = 47,248$ and unique TSS ±2 kbp (shaded area, lower panel), $n = 32,166$. **c** Genome tracks of H2A.Z ChIP-seq signal from different developmental stages and at corresponding input samples, see also Supplementary Fig. 1a. **d** Representative fluorescence microscopy images of P7, P10, and P12 oocytes stained with an H2A.Z antibody, α-H2A.Z (red) and Hoechst 33342 (blue), shown here with consistent brightness and contrast.

Bar = 20 μm. **e** Box plots of integrated fluorescence intensities in nuclei of P7($n = 13$), P10($n = 8$), and P12($n = 10$) oocytes. The lines inside the boxes represent the median, the boxes the interquartile range, where whiskers extend 1.5x of this range. $p$-values were calculated using Kruskal-Wallis testing ($p = 0.00003$) followed by Dunn for multiple comparisons with Benjamini-Hochberg corrections for multiple testing (* $p = 0.01$; **** $p = 0.00002$). Source data are provided as a Source Data file. **f** H2A.Z ChIP-seq peak overlap with genomic features in the mm10 genome, see also Supplementary Fig. 1f. **g** Principal component analysis of the intensity of H2A.Z enrichment at H2A.Z peaks at different stages.

stages, which had relatively lower library complexities. In contrast to prominent histone modifications, H2A.Z incorporation generally occurred in relatively narrow regions with a mean size of 877 bp. Annotation of these peaks showed that H2A.Z was frequently incorporated away from genes and TSSs (Fig. 1f). Specifically, only 27% of H2A.Z peaks were located near promoters, while 33% and 40% were intragenic and intergenic, respectively (Fig. 1f). To assess the overall changes and similarities in the H2A.Z enrichment profile, we performed general correlation analyses throughout the genome as well as Principal Component Analysis (PCA) of the signal at the aggregate set of H2A.Z peaks. This revealed considerable stage-specificity in the H2A.Z localization (Fig. 1g, Supplementary Fig. 2d), where embryos at the 8-cell, morula and blastocyst stages showed a high degree of similarity but were distinct from that of all oocyte stages. Likewise, oocytes at later stages of oogenesis (NSN, SN, MII) clustered together, demonstrating a high degree of similarity. Growing oocytes (P7, P10, and P12), however, were clearly distinct from NSN, SN, and MII oocytes. As growing oocytes are only separated by two to five days of growth, this supports a striking, rapid change in H2A.Z establishment early after oocytes enter the growth phase (Fig. 1g). After fertilization, H2A.Z enrichment gradually transitioned from a profile related to that of mature oocytes via a distinct path towards the later embryo profile.

To assess the chromatin context in which the oocyte and embryo H2A.Z signal exists, we made use of the mouse full-stack ChromHMM annotation[30]. This revealed enrichment of H2A.Z at regions characterized as TSSs, promoters, enhancers and open chromatin, and H2A.Z depletion in regions characterized as heterochromatin (Supplementary Fig. 2e). However, as these genomic states were defined using ChIP-seq data from somatic cells, the results mainly characterize the relationship to epigenomes of later developmental stages or adult tissues[1].

## H2A.Z is incorporated at CGIs near TSSs of expressed genes

To visualize H2A.Z enrichment at TSSs over time, we performed k-means clustering of the TSS-associated H2A.Z signal across developmental stages using TSSs defined from annotated genes in the mouse mm10 genome (Fig. 2a). Surprisingly, few clusters showed predominant enrichment in either the developing oocyte (cluster VII) or the early embryo (clusters V and VIII), and many clusters had persistent enrichment throughout all stages (clusters I–IV and VI). Approximately 40% of TSSs had negligible H2A.Z signal and these TSSs were also lacking CGIs (clusters IX and X) (Fig. 2a). Next, we related the clustered TSSs to previously published RNA expression levels encoded by nearby loci[31] and found an overall positive relationship between H2A.Z and RNA levels (Fig. 2b). Only one cluster (IV) displayed changing RNA levels, while the others were associated with constant RNA levels across the developmental stages. An investigation of developmentally regulated genes within the clusters showed no specific enrichment or depletion in the oocyte or early embryo clusters. However, clusters II and III with constant RNA expression were depleted of maternal-specific genes and enriched in genes transcribed during major ZGA and mid-preimplantation gene activation (MGA), while clusters IX and X with low H2A.Z enrichment and low gene expression displayed the opposite pattern (Fig. 2c). Minor ZGA genes

were mainly enriched in cluster IV, with corresponding high RNA levels at the oocyte and zygote stage and low RNA levels at later stages. Overall, H2A.Z was enriched at genes expressed at ZGA and MGA both during oogenesis and preimplantation embryo development, although the global levels of enrichment were lower at the 2-cell stage. Since we detected H2A.Z before the mouse minor and major ZGA, which occur in the zygote and the 2-cell embryo, respectively[23], H2A.Z may contribute to marking these genes for later transcription either directly or indirectly as a placeholder comparable to that reported in zebrafish and *Drosophila* embryos[32–34]. The relatively persistent H2A.Z enrichment observed at many TSSs during oogenesis and preimplantation development contrasts with the highly dynamic levels of many epigenetic marks at these stages[24]. This stability may reflect a role for H2A.Z as a placeholder during epigenetic restructuring, as previously suggested in zebrafish[4].

To determine whether H2A.Z enrichment was specifically associated with TSSs, or alternatively with CGIs at or proximal to TSSs, we calculated the proximity of TSSs to CGIs and sorted them accordingly. It was clear that the majority of the H2A.Z signal was located at CGIs (Fig. 2d, e, Supplementary Fig. 3a), whereas low-level enrichment was observed at TSSs that were not associated with CGIs (faint vertical line in Fig. 2d, Supplementary Fig. 3a and blue lines in Fig. 2e). When visualizing CGIs, gene bodies, GC density and CpG density (Fig. 2d), we found that most of the H2A.Z signal at TSSs not associated with CGIs was instead enriched in CpG-dense regions that were not part of annotated CGIs. Moreover, a positive correlation was found between CpG density and H2A.Z enrichment at TSSs for P12 onwards (Spearman's $\rho > 0.70$, Supplementary Fig. 3b), except for 2-cell, while the correlation was negligible in genome-wide assessments (Supplementary Fig. 3c). An analysis of the standard deviation of the H2A.Z signal across the oocyte and embryo stages showed that most of the variation in H2A.Z enrichment between the stages appeared at non-TSSs loci (Supplementary Fig. 3d). In summary, H2A.Z at TSSs is strongly associated with CGIs, and to a lesser extent, CpG-dense regions, while in other parts of the genome, H2A.Z exhibits a more variable distribution considerably less linked to CpG density.

## Dynamic H2A.Z signals reflect epigenetic changes

Since CGIs near TSSs can only explain a minority of the H2A.Z localization (Fig. 1f), and the most variable signal exists elsewhere in the genome (Supplementary Fig. 3d), we did an unbiased analysis of the full set of H2A.Z peaks from oocyte and embryo stages. To this end, we performed k-means clustering of the H2A.Z signal at all H2A.Z peaks across stages (Fig. 3a), except for the zygote to 4-cell stages, which had relatively lower library complexities compared to the other samples. This analysis revealed a clear pattern of stage-specific differences, with some clusters of H2A.Z enrichment being specific to early growing oocytes (cluster 1 and 2), further progressed, fully mature oocytes and zygotes (cluster 3), as well as later embryonic stages (cluster 8-10). Additionally, four clusters displayed a more persistent H2A.Z enrichment from P12 onwards (clusters 4–7), and these clusters were strongly enriched for CpGs in comparison to the genome-wide mean (Fig. 3a). Several related developmental stages showed high levels of similarity, as also observed in the PCA (Fig. 1g), demonstrating consistency in our

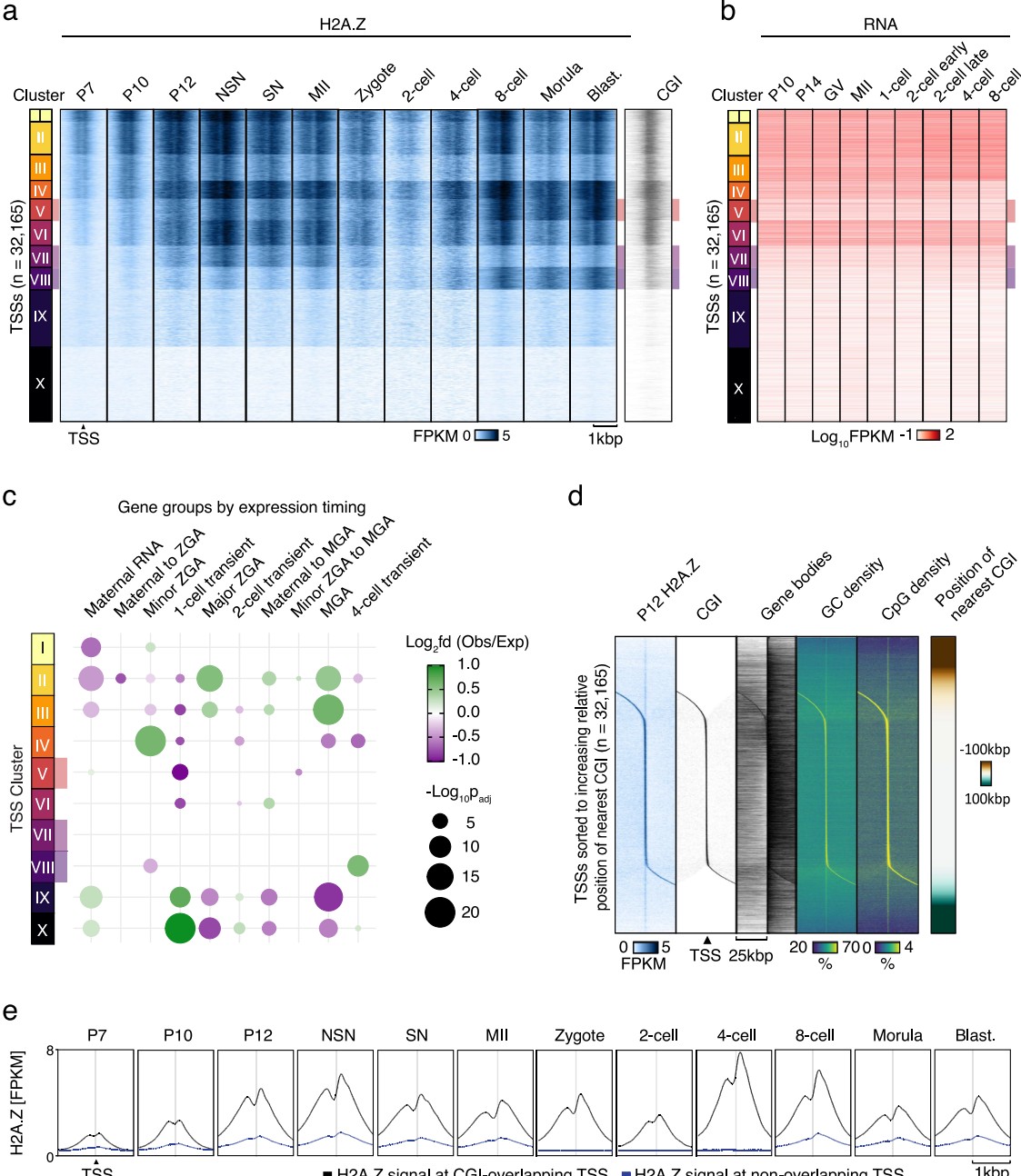

**Fig. 2 | H2A.Z incorporation at TSSs of developmental genes and CGIs. a** H2A.Z ChIP-seq signal in different developmental stages at unique TSSs ±1 kbp (*n* = 32,166), clustered based on H2A.Z signal. Blast., blastocyst. **b** RNA levels[31] at unique TSSs ±500 bp (*n* = 32,166), clusters based on Fig. 2a. **c** Bubble plots showing overlap between TSS clusters and oocyte and embryonic gene categories compared to an average distribution across all clusters. *p*-values were calculated by two-sided Chi-Square tests, Benjamini-Hochberg adjusted for multiple testing.

Bubbles corresponding to a *p*-value < 0.05 were excluded from the plot. ZGA, zygotic genome activation; MGA, mid-preimplantation gene activation. **d** Heatmaps showing H2A.Z, CGIs, Gene bodies, G/C, and CpG densities at unique TSSs ±25 kbp (*n* = 32,166) sorted according to increasing relative position of the nearest CGI. **e** H2A.Z in different developmental stages at unique TSSs (*n* = 32,166), either overlapping (black) or not overlapping (blue) CGIs. Blast., blastocyst.

data. As a control measure, we assessed the pattern of H2A.Z signal in four biological replicates of P10 oocytes from two different mouse strains, and found it to be highly similar (Supplementary Fig. 4a). Likewise, H2A.Z enrichment patterns were similar between the blastocyst sample pool derived from this study and previous data[15] (Supplementary Fig. 4b). Altogether, these findings support the specificity of H2A.Z enrichment observed across developmental stages.

In most cell types, the H3K4me3 histone modification is associated with active transcription and open chromatin, while H3K27me3 is associated with repressive chromatin and transcriptional silencing,

and both marks are localized at CGIs[1]. During oogenesis and pre-implantation embryo development, however, the distribution of H3K4me3[31,35] and H3K27me3[36] becomes atypical and organized into broad domains. To study incorporation of H2A.Z in the context of these extensive epigenetic changes as well as chromatin accessibility, we assessed association between clusters of H2A.Z and profiling of H3K4me3[35] and H3K27me3[36] as well as previously published ATAC-seq[37-39] in mouse oocytes and embryos (Fig. 3b). The dynamic H2A.Z signal was mirrored by changes in the epigenetic context, and we observed a strong stage-specific enrichment of H3K4me3 and open

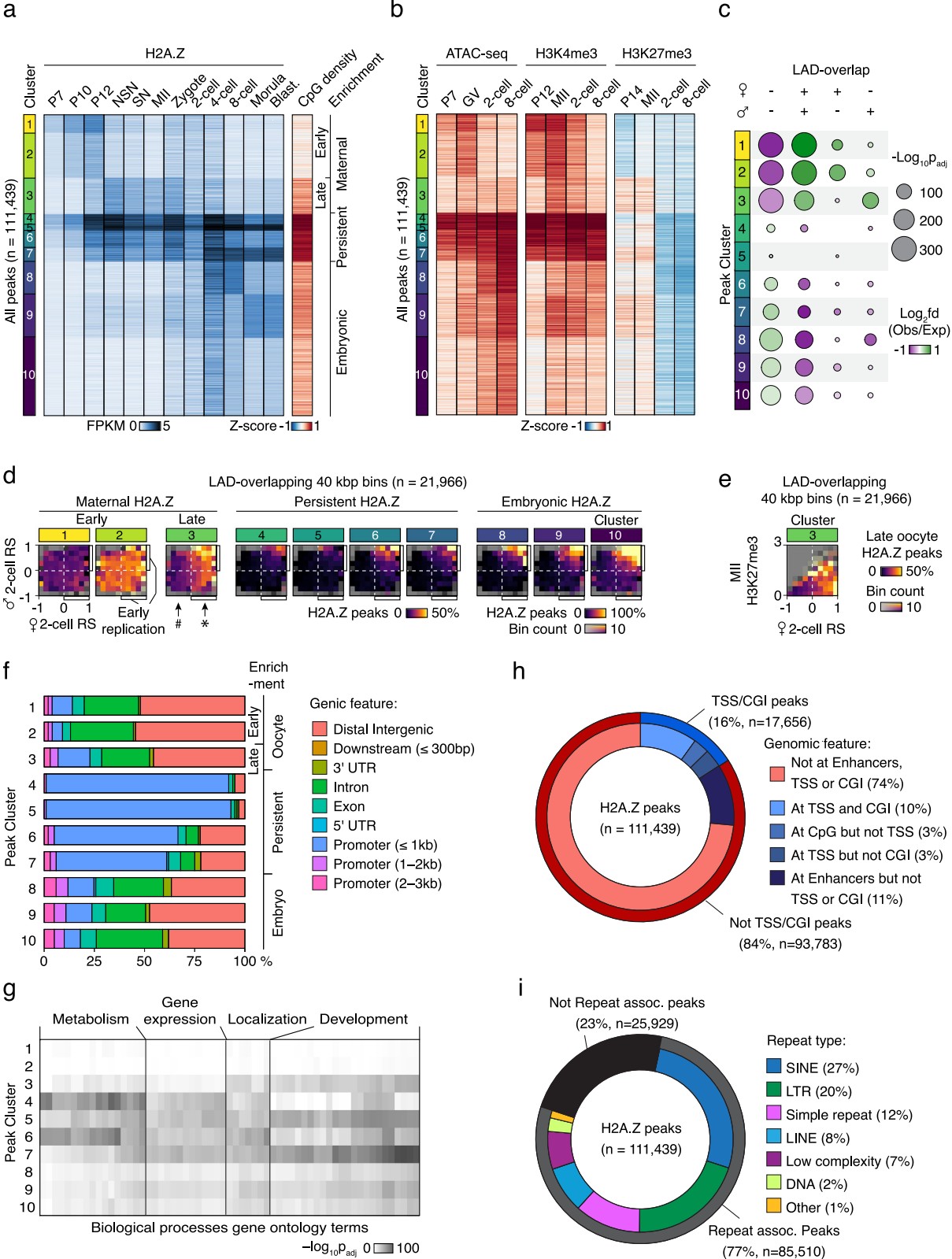

chromatin at H2A.Z peaks, while H3K27me3 was predominantly depleted, particularly in 8-cell embryos (Fig. 3a, b). Notably, early maternal H2A.Z peaks (clusters 1 and 2) displayed high to intermediate enrichment of H3K4me3 and open chromatin in P12 compared to the mean genome-wide signal, while P14 H3K27me3 was depleted at these peaks (Fig. 3b). The P12 pattern persisted to some degree to the MII stage, although both H3K4me3 and H3K27me3 exist as broad domains

at this time. Intriguingly, accessibility at peaks in the early maternal clusters (1 and 2) remained elevated in GV oocytes at a stage where H2A.Z levels were reduced, which might be linked to the appearance of broad non-canonical H3K4me3. H2A.Z and H3K4me3 enrichment at the 2-cell stage both appeared to be transitioning towards the pattern observed in 4-cell and 8-cell stages, while H3K27me3 already had a similar pattern to the 8-cell stage at this time. At the 8-cell stage, the

**Fig. 3 | Oocyte and embryo specific H2A.Z peaks are mainly found outside of genes. a** Heatmaps of H2A.Z enrichment at aggregated and clustered H2A.Z peaks from all developmental stages, except for the zygote to 4-cell stages, which had relatively lower library complexities compared to the other samples. Z-score of CpG density at peaks is also shown. Blast., blastocyst. **b** Z-scores of histone marks[35,36,71] and ATAC-seq[37–39] at clustered peaks. **c** Bubble plots showing overlap of LADs[22] and clusters of H2A.Z peaks compared to an average distribution across all clusters. *p*-values were calculated by two-sided Chi-Square tests, Benjamini-Hochberg adjusted for multiple testing. Bubbles corresponding to a *p*-value < 0.05 were excluded from the plot. **d** 2D-histograms showing the genome-wide occurrence of each cluster of H2A.Z peaks (color) in relation to the mean Replication Status (RS) of the maternal (X-axis) and paternal (Y-axis) genomes[25] in individual 2-cell embryos. # and * indicates noteworthy sex-specific differences, where low and high peak-occurrences, respectively, largely follow the maternal RS, but not the paternal. Data was analyzed in 40 kbp bins, and the subset of bins overlapping with

LADs is shown (*n* = 21,966). For the non-overlapping subset, see Supplementary Fig. 4d. **e** 2D-histogram showing the genome-wide occurrence of cluster 3 H2A.Z peaks (color) in relation to the mean maternal RS (X-axis) in individual 2-cell embryos and H3K27me3 levels in MII oocytes (Y-axis). Data was analyzed in 40 kbp bins, and the subset of bins overlapping with LADs is shown (*n* = 21,966). For the non-overlapping subset, see Supplementary Fig. 4d. **f** Overlap between peaks in each cluster and indicated genic features. **g** Heatmap of gene ontology (GO) analysis for H2A.Z peak annotated genes. -log$_{10}$ transformed adjusted *p*-values of gene ontology biological process terms (X-axis) are visualized across H2A.Z peak clusters (Y-axis). Biological processes with a cumulative-log$_{10}$ adjusted *p*-value in all clusters greater than 150 are visualized and labeled as general biological processes for clarity. One-tailed Fisher's exact test was used to calculate the *p*-values, followed by multiple testing correction with g:SCS. **h** Overview of the fraction of peaks that overlap with genomic features. **i** Overview of the fraction of peaks that overlap with major classes of repetitive elements. Assoc., associated.

---

H3K4me3 and ATAC-seq signals were reduced at early maternal peaks, while the H3K27me3 level was depleted in all H2A.Z clusters. The persistent H2A.Z enrichment in clusters 4−7 was characterized by open chromatin and strong enrichment of oocyte H3K4me3 as well as CpGs at all analyzed stages (Fig. 3a, b). Concordantly, clusters 4-7 were largely depleted when removing peaks overlapping with TSSs or CGIs from the peak set, (Supplementary Fig. 4c). While H2A.Z signal intensity at H2A.Z peaks was positively correlated to H3K4me3 levels in P12 oocytes and 8-cell embryos at TSSs (Spearman's ρ = 0.83), correlations were reduced in the rest of the genome and at TSSs in MII oocytes (Supplementary Fig. 4d). In conclusion, this suggests a closer relationship between H3K4me3 enrichment compared to H3K27me3 enrichment for the dynamic H2A.Z signal.

## H2A.Z marks early replication timing and LADs
Nuclear organization is established de novo shortly after fertilization with the formation of LADs, which are often related to gene repression[22]. Interestingly, the LAD formation of the two parental genomes differs until the 8-cell stage[22]. We first visualized the overall relationship between LADs and H2A.Z peaks and found that both maternal, paternal, and shared LADs in 2-cell embryos had a markedly lower density of H2A.Z peaks (Supplementary Fig. 4e). To investigate if oocyte H2A.Z is associated with LAD formation, we next analyzed the level of overlap between clustered H2A.Z peaks and LADs that were specific or shared between the two parental genomes in 2-cell embryos (Fig. 3c). We observed a highly significant co-occurrence between H2A.Z peaks arising in mature oocytes (cluster 3) and LADs that were specific for the paternal genome but not the maternal genome. As maternal H2A.Z peaks are maternally inherited, this indicates that factors in this genome and/or chromatin counteracted LAD formation (Fig. 3c). This was seemingly unrelated to the presence of broad H3K4me3 domains in the MII oocytes, since early oocyte H2A.Z peaks in cluster 1 and 2 underwent highly significant LAD-formation in the maternal genome only, despite being characterized by abundant MII H3K4me3 (Fig. 3c).

Recent investigations of replication timing in mouse early embryos and 2-cell-like cells revealed a slower DNA replication up until the 8-cell stage and the gradual emergence of a timing program in 2-cell embryos that gets progressively stronger until the 8-cell stage[25,40–43]. The timing program correlates with transcription, LADs, genome compartmentalization and inherited histone modifications with consistent differences between the parental genomes[25,41,42]. Given the involvement of H2A.Z in the activation of replication origins and replication timing[5], we next investigated this in our data. As LADs and late replication are strongly correlated[25], we decided to analyze the parts of the genome overlapping and not overlapping with maternal 2-cell LADs separately (Fig. 3d, Supplementary Fig. 4f). When relating the occurrence of the different developmental stage-specific H2A.Z peaks to the replication timing of the maternal and paternal genomes

in the 2-cell embryo, we discovered a clear relationship between the maternal replication timing and the H2A.Z peaks occurring in mature oocytes (cluster 3) (Fig. 3d, maternal late and early replication indicated by # and *, respectively). In contrast, no such relationship was seen for early maternal H2A.Z peaks (clusters 1 and 2) or for the paternal genome (Fig. 3d). Also, peaks from clusters with more persistent H2A.Z enrichment (clusters 4−7) or embryonic H2A.Z (clusters 8−10), were largely associated with the early replication of both the maternal and paternal genomes. The asynchronous early 2-cell replication of the maternal genome is defined by maternally inherited MII oocyte H3K27me3[25], which recently was shown to antagonize LAD formation in the embryo[44]. Accordingly, we investigated whether the mature maternal H2A.Z incorporation was linked to asynchronous early maternal 2-cell replication through H3K27me3. While maternal H2A.Z peaks were frequent in such genomic regions characterized by H3K27me3 in MII oocytes, these peaks were also observed in counterparts with little or no preceding H3K27me3 in MII oocytes (Fig. 3e, Supplementary Fig. 4g). We furthermore used previously published replication timing data[42] to analyze the relationship between replication timing in 4-cell and 8-cell embryos and the occurrence of the different clusters of H2A.Z peaks (Supplementary Fig. 4h, i). These analyses showed that both the clusters with embryonic H2A.Z enrichment (clusters 8-10) and the clusters with persistent H2A.Z enrichment (clusters 4−7) were associated with early replication timing of 4-, 8-, and 16-cell embryos (Supplementary Fig. 4h, i) and were not generally present within parts of the genome replicated early in 2-cell embryos (Supplementary Fig. 4h). Conversely, peaks from the cluster characterized by late maternal H2A.Z (cluster 3) were more likely in parts of the genome that were replicated early in 2-cell embryos and less likely in 4-cell embryos (Supplementary Fig. 4h). Altogether, these intriguing relationships imply that maternally inherited H2A.Z incorporation or associated features may be instructive for LAD formation and replication timing in the embryo through a mechanism independent of H3K27me3.

## Dynamic H2A.Z signals are intergenic and repeat-associated
To characterize stage-specific and persistent H2A.Z enrichment further, we analyzed the occurrence of genic features for each cluster. We found that the majority of the oocyte and embryo stage-specific H2A.Z incorporation took place at distal intergenic and intronic regions, while the clusters 4−7 with persisting H2A.Z enrichment were strongly enriched in promoters (Fig. 3f). To explore potential biological implications of H2A.Z incorporation, we next annotated peaks to TSSs of genes within 5 kbp of them and performed a gene ontology analysis for each peak cluster (Fig. 3g, Supplementary Data 3). We found an overall enrichment of H2A.Z peaks at genes frequently involved in metabolism, gene expression, localization, and development-related biological processes. Intriguingly, the early maternal clusters (1 and 2) did not show any strong enrichment, while the late maternal cluster 3 showed

an enrichment for localization and development-related genes, but limited association to biological processes related to gene expression. This analysis also revealed two different subsets of the clusters with more persisting enrichment, namely clusters 4 and 6, contrasting to cluster 5 and 7. While clusters 4 and 6 showed a strong enrichment in genes involved in metabolism-related processes, clusters 5 and 7 were enriched in genes involved in development-related processes. Taken together, this analysis highlights that both stage-specific and persistent H2A.Z enrichment are coupled to the biological context of nearby genes involved in oogenesis and early embryonic development.

We next conducted a comprehensive analysis of genomic features at or near non-TSS/CGI H2A.Z peaks and confirmed the enrichment of H2A.Z at enhancers as well as in open chromatin regions (Supplementary Fig. 5a, b). At enhancers[39,45], the signals were highly dynamic, indicating involvement in gene regulation (Supplementary Fig. 5a). The most notable changes at enhancers occurred between early growing oocytes, further progressed oocytes, and the embryonic stages from 4-cell to blastocyst, with zygote and 2-cell being intermediate, and enrichment was highly similar within each of these groups. Moreover, non-TSS/CGI H2A.Z peaks were often associated with genes involved in transcriptional regulation and cell morphology (Supplementary Fig. 5c). While 16% of H2A.Z peaks overlapped with TSS or CGIs and 11% with general enhancers, 74% were located in other genomic regions (Fig. 3h). To identify possible features shared by the large fraction of H2A.Z peaks that did not overlap with TSSs and CGIs, we analyzed the overlap between this subset and a repeat annotation from Repeatmasker[46]. We found that 77% of H2A.Z peaks colocalized with repetitive elements, mainly SINE and LTR TEs (Fig. 3i). Only 18% of H2A.Z peaks did not colocalize with either a repetitive element, a general enhancer, a TSS or a CGI. Thus, the majority of H2A.Z deposition in the genome of mouse oocytes and early embryos occurs at TEs and other repetitive elements.

## Stage-specific H2A.Z at LTR retrotransposons

Given that TEs are expressed in stage-specific waves in mammalian embryos, with specific TEs being essential for embryonic development[19], we investigated which repetitive elements were enriched in our stage-specific H2A.Z peak clusters (Fig. 4a, b). The most abundant repetitive elements were long terminal repeat (LTR) RTs, particularly of the RMER and MT types. These were specific for H2A.Z established in early growing oocytes (clusters 1 and 2) (Figs. 4b, S5a), indicating that the major wave of H2A.Z taking place at this stage is associated with RTs. The H2A.Z specifically incorporated during later oocyte stages (cluster 3), occurred in parts of the genome defined by simple repeats and low complexity regions (Fig. 4b, Supplementary Fig. 6a). To explore the underlying DNA sequences, we performed de novo motif enrichment analysis of clustered non-TSS/CGI peaks and found an increased frequency of GC rich motifs in peaks belonging to clusters 3, 8, 9 and 10 (Supplementary Fig. 6b). Furthermore, the long C-rich motifs identified in clusters 3 and 9 were highly specific for these clusters despite considerable similarity (Supplementary Fig. 6c). A three-way analysis of the genome-wide relationship between H2A.Z, CpG densities, and GC% revealed that H2A.Z in general was largely associated with CpG density and not GC%, evident from the high GC% regions with low CpG density having lower H2A.Z signals than high CpG regions (Supplementary Fig. 6d). These findings suggest that the association of H2A.Z to certain repetitive elements may be due to elevated CpG densities within repetitive elements rather than specific motifs.

The high copy number of TEs provided an ideal basis for systematically probing relationships between H2A.Z, histone marks, and transcription. We therefore investigated the localization and dynamics of H2A.Z in selected populations of enriched TE types and observed a localized enrichment near a considerable fraction of RMERs and MTs (Fig. 4b, c). Indeed, a large fraction of MTAs, which is the youngest and most abundant MT subfamily[47,48], were associated with dynamic H2A.Z enrichment across oocyte and embryo stages co-occurring with open chromatin (Fig. 4c, Supplementary Fig. 7a). H2A.Z levels differed in timing and localization relative to the MTAs (Supplementary Fig. 7a), with the most pronounced and proximal enrichment occurring early in oocyte development and peaking at the P12 stage (Supplementary Fig. 7a, see *). Despite the many distinct H2A.Z profiles associated with MTAs, the relationship between H2A.Z profiles and transcription was limited, as many H2A.Z-negative MTAs were transcribed at the same level as H2A.Z-positive MTAs (Supplementary Fig. 7a). In the P12 stage oocytes, increased upstream H2A.Z enrichment was associated with reduced upstream expression, suggesting that H2A.Z may strengthen transcriptional directionality at MTAs here (Supplementary Fig. 7a, see *). These findings indicate an intriguing relationship between active RTs and H2A.Z establishment in growing oocytes, where the strongest H2A.Z enrichment is associated with the absence of transcription.

Mammalian oocytes have a distinct transcriptome with an abundance of LTR-initiated transcription units (LITs), which are genic and intergenic transcripts initiating from solo LTRs[47,48]. Of the 3384 LITs previously identified in the mouse genome, many are MTs ($n = 2294$), especially MTAs ($n = 1345$). When visualizing H2A.Z specifically at LITs, a narrow and localized H2A.Z signal was generally evident at the P10 and P12 stages (Fig. 4d). More than half of these LITs (2165) did not overlap with detected H2A.Z peaks (Fig. 4d), but a low level of sub-threshold H2A.Z signal was still evident. Strong H3K4me3 enrichment at LITs in P12 and P15 oocytes showed that this epigenetic mark coexists with H2A.Z at a considerable fraction of these elements (Fig. 4e). Likewise, the LITs with H2A.Z signal at CGIs (237) (Fig. 4d) displayed both H2A.Z and H3K4me3 enrichment during early oogenesis and embryo development, only interrupted at the MII stage by the formation of broad H3K4me3 domains and with a slightly lower signal at the 2-cell stage. In contrast, the H3K27me3 signal remained low within LITs (Fig. 4e). These findings show a dynamic pattern of H2A.Z and H3K4me3 at LITs with a considerable degree of colocalization.

## H3K4me3-independent H2A.Z is incorporated into low-CpG loci

In contrast to TSSs, where we found a strong correlation between H2A.Z and H3K4me3, the strength of the association between H2A.Z and H3K4me3 was reduced at the genome-wide level (Supplementary Fig. 4d). To investigate this in greater detail, we related genome-wide H2A.Z and H3K4me3 signal in P12, MII and 8-cell stages to CpG enrichment (Fig. 5a). Interestingly, areas of the genome with the lowest levels of H2A.Z and H3K4me3 coexistence were also characterized by low CpG density in P12, MII and 8-cell stages, whereas coexistence was generally associated with high CpG densities (Fig. 5a). To probe these intriguing relationships further, we again took advantage of the high copy numbers of RTs and clustered selected RTs based on the combined profiles of H2A.Z, H3K4me3 and H3K27me3. Visualization of common epigenetic marks, CpG and GC densities, as well as chromatin accessibility at the clustered RTs in growing oocytes, revealed that regions with high levels of either H2A.Z or H3K4me3 were low in H3K27me3 and DNA methylation, and vice versa (Fig. 5b, Supplementary Fig. 7b). Furthermore, H2A.Z and H3K4me3 had minimal colocalization, especially at MTA RTs (Fig. 5b, Supplementary Fig. 7b), where H2A.Z generally occurred more strongly next to the start sites of MTA RTs with high accessibility. This enrichment in ATAC-seq signal, combined with the centrally localized absence of H2A.Z signal directly at the MTA start sites, might reflect nucleosome-free regions. While both H2A.Z and H3K4me3 signals were generally present at the same loci, a notable spatial separation existed with H2A.Z mainly situated upstream of H3K4me3 (Fig. 5b, c, clusters E, G and J). Moreover, H3K4me3 and RNA levels tended to be associated with an elevated abundance of CpG and GC at the regions flanking these MTA subgroups. H3K4me3 was largely absent in the approximately 30% of MTA

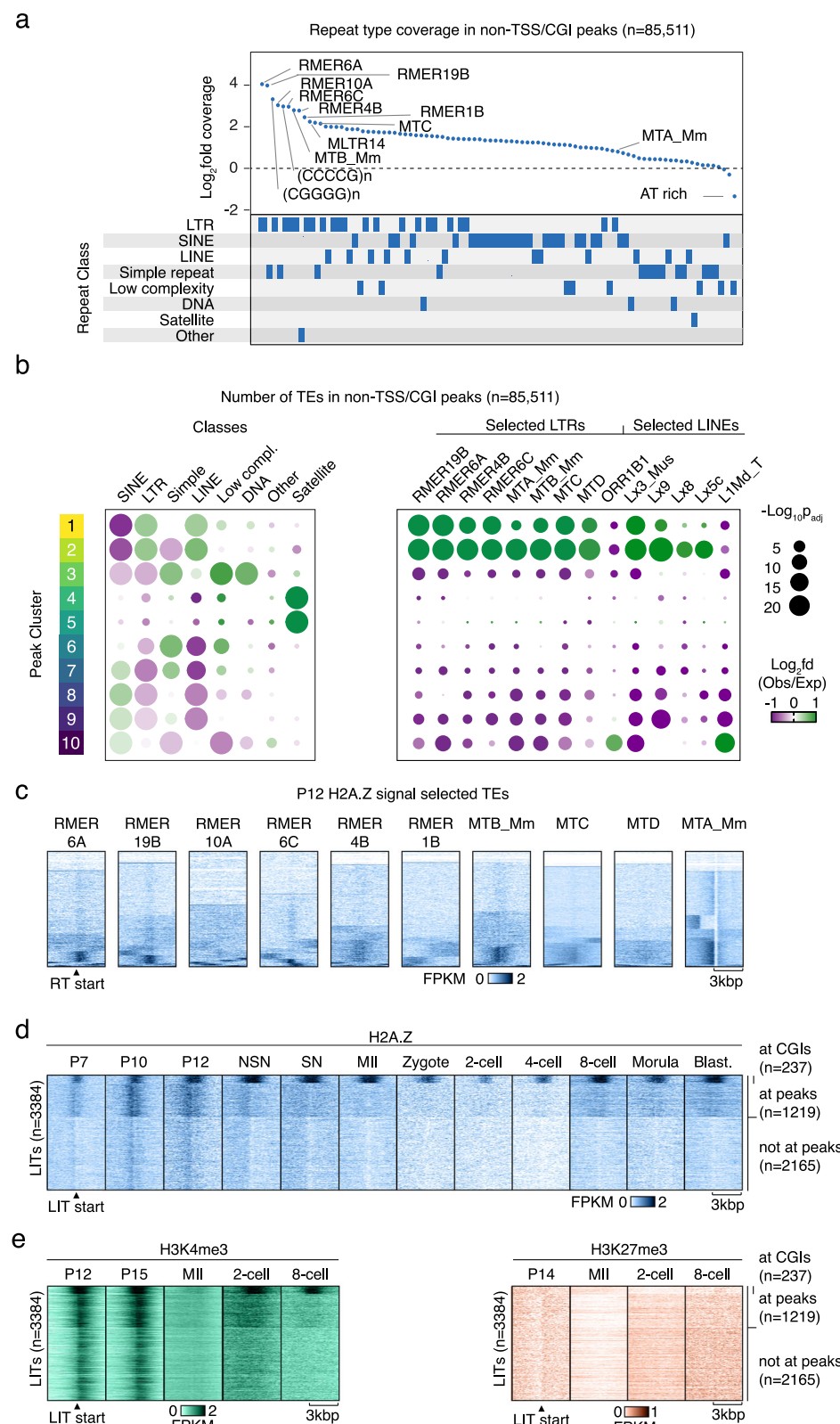

**Fig. 4 | Maternal H2A.Z signal is enriched at specific repetitive elements.**
**a** Log2fold difference between observed and expected coverage of 90 repetitive elements with >200 counts in the non-TSS/CGI H2A.Z peak set. The bottom panel shows the repeat class. **b** Bubble plots showing overlap of TE classes (left) and selected LTR and LINE subtypes (right) in clusters of non-TSS/CGI H2A.Z peaks compared to an average distribution across all clusters. *p*-values were calculated by two-sided Chi-Square tests, Benjamini-Hochberg adjusted for multiple testing.

Bubbles corresponding to a *p*-value < 0.05 were excluded from the plot. A complete plot of all TE types can be found in Supplementary Fig. 6a. **c** Heatmaps showing H2A.Z enrichment at selected RTs during the P12 stage clustered based on the H2A.Z profile at the start of the RTs ±3 kbp. **d** Heatmaps of H2A.Z enrichment profiles at the start of LITs[48] ±3 kbp sub grouped based on overlap to CGIs and H2A.Z Peaks. Blast., blastocyst. **e** H3K4me3[35] and H3K27me3[36,71] profiles at LITs ordered as Fig. 4d.

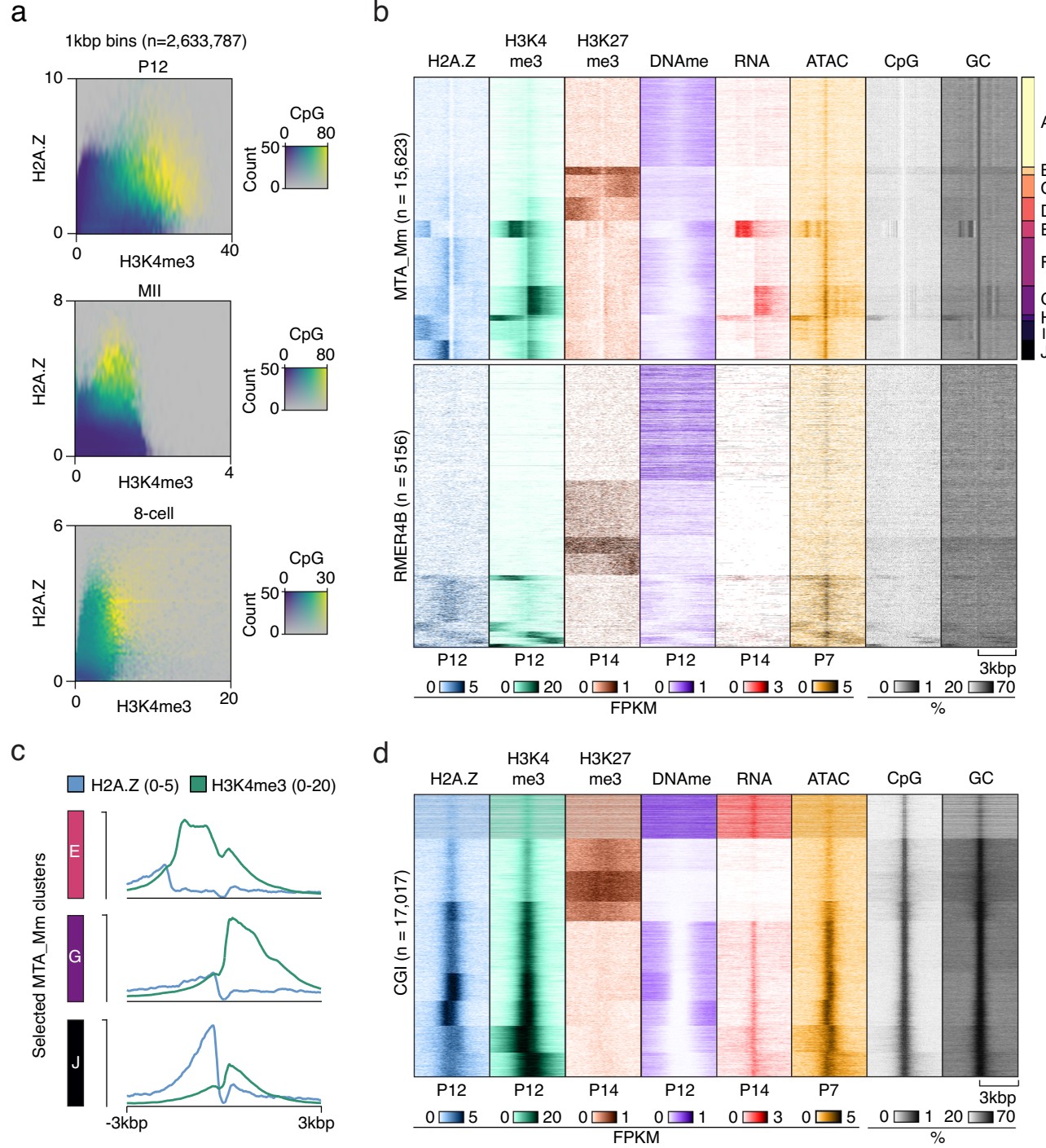

**Fig. 5 | H2A.Z signal is not correlated to H3K4me3 levels and CpG density at many loci. a** Heatmaps showing the genome-wide relationships between H2A.Z, H3K4me3[35] and CpG density in selected developmental stages visualized in 1 kbp bins (n = 2,633,787). Opacity reflects the occurrences of indicated H3K4me3 and H2A.Z levels. **b** Heatmaps showing indicated features[31,39,72] at selected RTs clustered based on the combined distribution of H2A.Z, H3K4me3[35] and H3K27me3[71] at the start of each RT ±3 kbp in growing oocytes (P12 or P14). **c** Average H2A.Z and H3K4me3 signal from selected clusters from 5b. **d** Heatmaps showing indicated features[31,39,72] at CGIs clustered based on the combined distribution of H2A.Z, H3K4me3[35], and H3K27me3[71] at the start of each CGI in growing oocytes (P12 or P14).

loci displaying high H2A.Z signal (Fig. 5c, clusters F and I), and only the minor cluster H showed colocalization of H3K4me3 and H2A.Z. Increased RNA levels were mainly observed at loci in clusters E and G, generally in open chromatin enriched in H3K4me3, and a similar trend was observed for MTB (Supplementary Fig. 7b). When examining older types of RTs such as MTC and MTDs[47], a greater fraction of the H2A.Z signal overlapped with H3K4me3 (data not shown), suggesting that

lack of overlap between H2A.Z and H3K4me3 is a feature of younger types of MT RTs. In contrast, RMERs, Lx3_Mus and Lx9 LINE RTs demonstrated more widespread overlap of H2A.Z and H3K4me3 signals (Fig. 5b, Supplementary Fig. 7b), indicating a different regulatory mechanism for MTA and MTB RTs. In comparison, H2A.Z and H3K4me3 generally coexisted at CGIs (Fig. 5d), altogether demonstrating the existence of an intricate sequence-dependent

relationship between the histone variant H2A.Z and the histone modification H3K4me3.

## Discussion

In this study, we provide genome-wide maps of the histone variant H2A.Z during six different stages of mouse oogenesis, as well as six stages of preimplantation development. Our work demonstrates a rapid major wave of establishment of H2A.Z in early oogenesis, peaking in P12 and NSN oocytes just prior to global transcriptional silencing. The landscape of H2A.Z signals is generally comprised of narrow peak profiles and enrichment at TSSs and CGIs. Thus, the profile of the histone variant H2A.Z differs markedly from previously assessed epigenetic marks at these developmental stages, including H3K4me3, H3K27me3, and DNA-methylation, which all display broader or otherwise non-canonical distributions that are unique for oocytes and preimplantation embryos[24]. This could allow epigenetic information to be preserved across these developmental stages despite major concurrent changes in H3K4me3, H3K27me3, and DNA methylation. H2A.Z may play a crucial role in the regulatory dynamics, potentially acting as a placeholder that directs where future epigenetic marks should be deposited during the restructuring of the epigenome. Indeed, such a function has been proposed in zebrafish and *Drosophila* embryos, where H2A.Z is essential for ZGA and is detected at genes that will later become active or poised[32–34]. This function may be particularly important for the H2A.Z-enriched loci that characterize the minor ZGA, major ZGA, MGA, and the MT subfamily of RTs observed in our study, as these loci all become active in the zygote or shortly thereafter.

During manuscript revision, additional studies addressing H2A.Z enrichment in mouse oocytes were published[27,28]. Although their focus is distinct from ours, these studies also show high H2A.Z enrichment early in oocyte development, contrary to earlier studies. Furthermore, their analyses corroborate our observed correlation between H2A.Z and H3K4me3 at TSSs and anticorrelation to DNA methylation. Additionally, these two studies include knockout models showing that H2A.Z is indispensable for oocyte maturation.

Apart from the fairly stable H2A.Z enrichment at TSSs from P12 onwards, our findings also reveal dynamic, stage-specific changes in H2A.Z profiles during oogenesis and preimplantation embryo development (Fig. 6a). Intriguingly, the maternal H2A.Z signal was strongly associated with absence of LADs and early replication timing in the maternal genome, indicating that maternal H2A.Z incorporation or related features may be instructive for the replication timing in the embryo. Additionally, these dynamic changes are particularly striking at non-TSS loci with low CpG enrichment, which often harbor TEs. There is a growing evidence that RT TEs are intimately intertwined with mammalian embryonic development (reviewed in[49]). For example, the LINE family of RTs is transcriptionally active early in preimplantation development, peaking at the 2-cell stage in mouse[50]. Furthermore, H2A.Z is enriched at LINE L1 promoter regions in mouse cell lines, possibly associated with silencing of these young RTs[51], while young RTs were also shown to be marked by H2A.Z in zebrafish embryos[52]. In our study, we identify several LINEs enriched for H2A.Z in the early maternal clusters (Supplementary Fig. 6a), particularly Lx3_Mus and Lx9, with lower enrichment observed for Lx5c and Lx8. However, Lx RTs are older LINEs[53], and we observed little accumulation of transcripts at these loci (Supplementary Fig. 7b). We also observed H2A.Z signal at RMERs and MTs (Fig. 4b), which are crucial for regulating the oocyte transcriptome and epigenome, and are frequently found in TE-initiated transcripts in mouse embryos[16,47,48]. The MT subfamily accounts for 13% of all transcripts in the fully grown oocyte[54] and has been co-opted to drive expression during ZGA and early embryo development[37,55]. The incorporation of H2A.Z at these RT TEs, having a role in embryonic development, further indicates that H2A.Z may be involved in the regulation of these critical developmental stages.

The non-TSS loci with H2A.Z deposition at RTs exhibited rather distinct and sometimes intriguing patterns of colocalization with H3K4me3: some followed a canonical pattern with H2A.Z and H3K4me3 overlap, some displayed H2A.Z positioned upstream of H3K4me3, and others lacked H3K4me3 entirely. While LITs were enriched in regions where H2A.Z overlapped with H3K4me3, an inverse relationship between H2A.Z and H3K4me3 was particularly clear in low CpG environments, such as at MTA and MTB RTs. The expressed MT RTs predominantly exhibited H3K4me3 downstream of H2A.Z (Fig. 5b, c, clusters E and G), except for a small subset displaying the canonical pattern (Fig. 5b, cluster H). These may correspond to the MTs previously shown to be active during early embryonic development[18]. Based on these observations, we propose a dynamic equilibrium model in which canonical H2A.Z is incorporated at CGIs that colocalize with H3K4me3 and active gene expression but is continuously evicted by transcription and replenished through recruitment from CGIs (Fig. 6b). In contrast, H2A.Z not overlapping with H3K4me3 is found in GC-poor regions upstream of H3K4me3, likely due to H2A.Z eviction at TSSs by active transcription and reduced incorporation rates caused by low CpG density (Fig. 6b). In these regions, H2A.Z may regulate the directionality of transcription.

As mentioned previously, the library complexity of the zygote to 4-cell stages is lower than that of other developmental stages in this study. This does not materially affect conclusions about enrichment profiles made at hundreds or thousands of loci, but reduces the signal strength at the single locus level, peak-calling strength, and adds stochastic variation to PCA analyses and clustering. Future samples with higher library complexities from these stages may therefore improve the understanding of the overall dynamics. Additionally, antibodies against H2A.Z do not distinguish between the two different, non-redundant isoforms of H2A.Z in mice. These isoforms may differ in their association with H3K4me3 and have been shown to be differentially expressed across tissues[3,7,56]. Accordingly, we cannot formally rule out the possibility that the observed differences in H2A.Z patterns are influenced by distinct functional roles of the two isoforms. Moreover, the antibody used in this study does not distinguish between different posttranslational modifications of H2A.Z, nor does it differentiate whether H2A.Z is incorporated into homotypic (H2A.Z/H2A.Z) or heterotypic (H2A/H2A.Z) nucleosomes, or whether it is combined with other canonical histones or their variants (e.g., H3.3). These combinations influence nucleosome stability, which in turn can impact the transcription of nearby genes[10,14,57,58].

H2A.Z is well known for its role in facilitating transcription initiation and accordingly has been shown to be more easily evicted from chromatin than H2A nucleosomes[10,59]. Furthermore, H2A.Z is probably preferentially incorporated into new nucleosomes at active genes with high nucleosome turnover, since H2A.Z is constitutively expressed while H2A is mainly expressed during the S phase[60]. The H3 variant H3.3 also accumulates at TSSs of active genes, and a nucleosome with both H2A.Z and H3.3 further destabilizes chromatin structure, and is thought to facilitate chromatin opening to enable gene expression[4,61–63]. However, H2A.Z also accumulates in heterochromatin and silenced genes, perhaps facilitated by H2A.Z tails interacting less with DNA and more with other nucleosomes compared to H2A, forming more compact chromatin[10,59]. Accordingly, H2A.Z is a versatile histone variant involved in many different regulatory and cellular processes, and its multifaceted character, especially during oocyte and embryo development, is highlighted by our study. Overall, the distinct patterns of H2A.Z incorporation observed at CGI-containing TSSs, CpG-depleted TEs, together with its association with LADs and maternal early replication timing, indicate that H2A.Z serves different functions across genomic regions and developmental stages. This may imply a role for H2A.Z in linking the regulation of TEs to the broader epigenetic program, particularly during oogenesis, thereby laying the foundation for successful embryonic development. By characterizing these dynamic changes in H2A.Z distribution, this work advances our

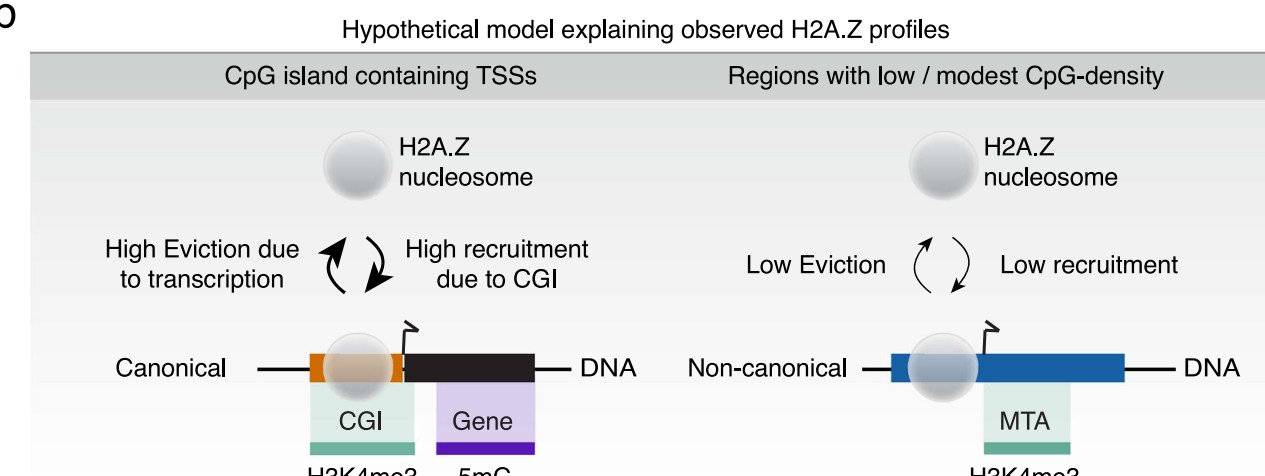

**Fig. 6 | Models of H2A.Z associated with histone marks, CGIs, TSSs and TEs. a** Model of H2A.Z in mouse oocytes and preimplantation embryos and its association with histone marks, CGIs, TSSs and TEs. **b** Hypothetical model of eviction and recruitment of H2A.Z due to CGIs.

understanding of how histone variants contribute to the establishment and maintenance of epigenetic landscapes in oocytes and pre-implantation embryos.

## Methods

### Mouse housing and care

Housing for the C57BL/6 NRj mice (Black 6N, Janvier Labs, France) and RjOrl:SWISS mice (CD-1®, Janvier Labs) was provided in individually ventilated cages (Green Line IVC SealSafe Plus Mouse, Tecniplast) connected to an Aero IVC air handling unit (with 75 air changes/hour), under specific pathogen-free conditions. The cages were equipped with Aspen bedding and wooden sticks (Tapvei, Estonia) and enriched with plastic shelters and paper wool nesting paper (Datesand, Manchester, GB). Standard housing conditions were maintained with $22 \pm 2\,°C$ room temperature and $55 \pm 5\%$ relative humidity, on a light:dark cycle of 12:12 h light (7 AM:7 PM), and free access to water and rodent chow. The staff of the animal facility at the Department of Comparative Medicine, Oslo University Hospital and the University of Oslo kept and cared for the mice.

### Mouse oocyte collection

**Strains and treatments.** Postnatal day 7 (P7), P10, and P12 CD-1 females were used for the collection of growing oocyte (GO) sample pools, as well as P10 Black 6 N females. Black 6 N females aged 4 weeks were used for collecting NSN, SN and MII oocytes (Supplementary Data 4). To induce superovulation for the collection of these three stages, 5 international units (IU) of pregnant mare serum gonadotropin (PMSG; Prospec, HOR-272) were injected intraperitoneally (IP) at 2 pm. For the NSN and SN stages, the oocytes were collected ~45 h after the PMSG injection. For the collection of MII stage oocytes, an additional intraperitoneal injection with 5 IU of human chorionic gonadotropin (hCG; Sigma, C1063) was given 45–48 h following the PMSG injection. Subsequently, MII oocytes were collected 18-22 h after the hCG injection.

**Mouse GO collection.** Dissected ovaries, 5 or 10 respectively, were placed in 3.5 mm dishes, and enzymatically digested in 0.08% Trypsin (Merck) at $37\,°C$ for 30 min. for ChIP-seq or in TrypLE Express (Gibco, 12605010) at $37\,°C$ for 15 min. for immunofluorescence, while the ovaries were punctured with a 30 G syringe (BD Micro-Fine + , Becton Dickinson). For ChIP-seq, this was followed by 20 min. treatment with 42 U/ml DNase (Sigma, 04716728001) at $37\,°C$ with 5% $CO_2$, then a 10 min. incubation with a final concentration of 1 mg/ml collagenase (Sigma, C9407) at $37\,°C$ with 5% $CO_2$ before collection, washing, and cross-linking as described below (picoChIP-seq > *Oocyte and embryo cross-linking*). Oocytes dissociated in TrypLE Express were

resuspended, and an equal volume of M2 medium was added for inactivation without further enzymatic treatment.

**Mouse NSN/SN oocyte collection.** Dissected ovaries with surrounding tissues were kept in M2 medium at 37 °C prior to processing, with approximately four organs being processed at a time. With the aid of a heated stage stereomicroscope (37 °C), fat and surrounding tissues were carefully removed, and the ovaries were punctured with a 30 G syringe (BD Micro-Fine + , Becton Dickinson) to mechanically release the oocytes into M2 medium supplemented with 0.2 mM 3-isobutyl-1-methylxanthine (IBMX; Sigma, I5879) to prevent meiotic resumption. Transfer to fresh, pre-warmed M2 medium with 0.2 mM IBMX was used for additional washes at 37 °C (no $CO_2$). The presence or absence of a perivitelline space (PVS), defined as a visible gap between the oolemma and the zona pellucida, was used to sort cumulus-free oocytes into NSN or SN stage. PVS serves as a reliable approximation for NSN/SN staging, as the ability to form a PVS within 1 h of in vitro culture with IBMX has been shown to correlate strongly with an SN chromatin configuration[64]. Finally, Tyrode's acid (Sigma, T1788) was employed to remove the zona pellucida, followed by washes in fresh M2 and cross-linking using formaldehyde, as described below (picoChIP-seq > *Oocyte and embryo cross-linking*).

**Mouse MII oocyte collection.** In a petri dish with M2 medium (Sigma, M7167), oviducts were transferred and examined using a stage stereomicroscope to identify the ampulla. Then, oocytes were mechanically released, and the surrounding cumulus cells were enzymatically removed using 0.3 mg/ml hyaluronidase (Merck, H3884) in M2, incubated at room temperature. Transfer to fresh M2 medium was used for additional washes. Finally, Tyrode's acid (Sigma, T1788) was employed to remove the zona pellucida, followed by M2 washes and cross-linking using formaldehyde, as described below (picoChIP-seq > *Oocyte and embryo cross-linking*).

**Superovulation, mating, IVF and mouse embryo collection**
**Zygote, 2-cell and 4-cell embryos.** To perform in vitro fertilization, sperm from C57BL/6NRj male mouse was collected directly from the cauda epididymis of a recently euthanized male and incubated in a 35 mm dish containing 90 μl of FERITUP preincubation media (Cosmo-Bio; KYD-002-EX). In parallel, a fertilization dish (30 mm) was prepared by placing a 90 μl droplet of human tubal fluid (HTF) medium supplemented with 2 mM reduced glutathione (Sigma; G4251), and covering it with NidOil. The HTF medium (100 ml) was prepared in-house (593.8 mg NaCl (Sigma; S-5886), 35 mg KCl (Sigma; P-5405), 4.9 mg $MgSO_4·7H_2O$ (Sigma; M-7774), 5.4 mg $KH_2PO_4$ (Sigma; P-5655), 57 mg $CaCl_2$ (Sigma; C-5670), 210 mg $NaHCO_3$ (Sigma; S-5761), 0.34 ml sodium lactate (Sigma; L-7900), 3.7 mg sodium pyruvate (Sigma; P-4562), 50 mg glucose (Sigma; G-6152), 49.55 mg Methyl-b-cyclodextrin (Sigma; C-4555), 0.5 ml penicillin/streptomycin (Gibco; 15140-122), 400 mg BSA (Sigma; A-7030), 0.04 ml phenol red (Sigma; P-0290), dissolved in Milli-Q water to a final volume of 100 ml). The fertilization dish was equilibrated at 37 °C in a 5% $CO_2$ incubator. Female C57BL/6NRj mice were superovulated via intraperitoneal injection of pregnant mare serum gonadotropin (PMSG; Prospec; HOR-272), followed 48 h later by human chorionic gonadotropin (hCG; Sigma; CG5). The dosage of each hormone (5 IU) was administered to mice aged 4 weeks. Fifteen hours after hCG administration, the mice were euthanized, and their oviducts were dissected and placed into the fertilization dish. Cumulus-oocyte complexes were retrieved from the ampulla and transferred into the fertilization droplet. Following a 30-min pre-incubation period, between 1 and 10 μl of motile sperm was added to the fertilization droplet. The dish was then incubated for three hours at 37 °C in 5% $CO_2$. To remove cumulus cells and excess sperm, embryos were washed through seven sequential droplets of Advanced KSOM medium (Sigma; MR-101) placed in a 60 mm dish, which had been pre-

equilibrated under the same conditions and covered with NidOil. Zygotes were collected approximately four hours post-fertilization, at which point pronuclei were clearly visible. Embryos at the 2-cell stage were collected 24 h post-fertilization, and those at the 4-cell stage were retrieved after ~48 h. All samples were subsequently processed for pico-ChIP sequencing as described below.

**Morulas and blastocysts.** Follicle growth was stimulated in 3.5–4 week old females (C57BL/6NRj, Janvier Labs) by an intraperitoneal (IP) injection with 5 IU PMSG at 2 pm. Around 45 h later, 5 IU hCG (Chorulon, MSD Animal Health) was IP injected to induce ovulation. Directly after hCG injection, females were housed 1:1 with stud males of the same strain for copulation. The following morning at 8am, females were removed from the males' cages and euthanized by cervical dislocation. The ovaries with oviducts and upper uterine horns were removed in one piece, placed in a microtube with 1 ml of M2 medium and immediately transferred for processing. Under a stereomicroscope, zygotes surrounded by cumulus cells were isolated from the ampulla in a petri dish with M2 medium at 37 °C. To remove the cumulus cells, zygotes were briefly put in hyaluronidase solution (HYASE-10X, Vitrolife) diluted 1:10 in M2 medium. The zygotes were then washed in three consecutive wells of 0.5 ml M2 medium covered with OVOIL-100 paraffin oil (Vitrolife). Normally fertilized zygotes, identified by the presence of two pronuclei, were selected and transferred to a pre-equilibrated micro-droplet culture dish (Vitrolife) containing 25 μl of G-1 PLUS culture medium (Vitrolife) per well, covered with paraffin oil, with approximately 10 zygotes per droplet. The zygotes were incubated at 37 °C with 6% $CO_2$. Late morulas ($n = 11$) and blastocysts ($n = 14$) were collected at developmental day 3.5, as embryos originated from natural copulation of several mice, and therefore development was not fully synchronized. The zona pellucida was removed using Tyrode's acid treatment, and the embryos were then washed in fresh M2 medium and cross-linked as described below (picoChIP-seq > *Oocyte and embryo cross-linking*).

**8-cell embryos.** The same procedure was followed for superovulation and copulation as described above, but with different timing for the hormone injections: PMSG at 3 pm and hCG 48 h later. At 68 h after the hCG injection, the females were euthanized by cervical dislocation and the ovaries with oviducts and upper uterine horns were removed and transferred for processing in a petri dish with M2 medium. One by one, the ovary was removed, the infundibulum was located and slid onto a flushing needle. Embryos were isolated by flushing M2 medium through the oviduct. 8-cell embryos were collected and washed in fresh M2 medium before Tyrode's acid treatment, further washing in M2, and cross-linking as described below (picoChIP-seq > *Oocyte and embryo cross-linking*).

**picoChIP-seq**
The picoChIP-seq protocol has been described in detail elsewhere (Manaf et al., manuscript in revision). A brief description of the different steps is provided:

**Oocyte and embryo cross-linking**
Oocytes were cross-linked in a solution of 50 μl M2 media, 50 μl PBS (Life Technologies, 14190-094), with a final concentration of 1% w/v formaldehyde (Sigma-Aldrich, F8775). After 8 min. at room temperature, 14.3 μl 1 M glycine (Merck, 67419) was added to inactivate the fixation. Following 5 min. at room temperature, ice-cold PBS was used to wash the oocytes three times in a 0.6 ml tube (Maximum Recovery, Axygen), resulting in a final volume of 10 μl. The fixative was added either in droplets under the microscope, or directly in the 0.6 ml tube, followed by the washes and centrifugations at 700 x $g$ for 10 min. at 4 °C with careful supernatant removal. Accordingly, the liquids were mixed by droplet pipetting or by gentle tube vortexing during cross-

linking and glycine inactivation. Liquid nitrogen was used to snap-freeze the samples prior to storage at −80 °C until further processing.

### ES cell cross-linking

The mouse E14 embryonic stem cells (ESC) were cultured as described in the mouse ENCODE project instructions, starting from a stock at passage P2, equivalent to the cells used for the landmark project (https://www.encodeproject.org/biosamples/ENCBS171HGC/). Prior to collection, cells were passaged once on gelatinized, feeder-free plates. Approximately 20 million cells in a pellet were cross-linked at room temperature for 8 min., by resuspension in 20 ml of 1% w/v formaldehyde (Sigma-Aldrich, F8775) solution in PBS, supplemented with 20 mM of sodium butyrate (Sigma-Aldrich, 19–137). The fixative was inactivated using 2.4 ml of an aqueous 1.25 M glycine (Sigma, G8790) solution, and the pellet was washed twice with ice-cold PBS supplemented with 20 mM of sodium butyrate (650 x $g$, 10 min., 4 °C). Following that, the cells were aliquoted into 0.6 ml tubes (Maximum Recovery, Axygen) at various cell numbers. Finally, the cells were centrifuged, the supernatant removed, leaving 10 μl, which was snap-frozen in liquid nitrogen and stored at −80 °C.

### Preparation of recombinant octamers

Plasmids for bacterial expression of recombinant human histones H2A, H2B, and H4 were generously provided by Robert Schneider (Helmholtz Zentrum, Munich, Germany) and H3.1 by Gunnar Schotta (LMU, Munich, Germany), and Histone octamers were prepared as reported previously[65]. The four core histones (H2A, H2B, H3 and H4) were expressed in *E. coli* BL21(DE3)pLysS, purified from inclusion bodies, and the pre-cleared extract was passed over a 5 ml HiTrap SP column in buffer (7 M Urea, 20 mM NaAcetate, pH 5.2, 200 mM NaCl, 1 mM EDTA, pH 8 and 5 mM b-mercapto-ethanol) and peak fractions dialyzed in SpectraPor 32 mm (MW 3000-6000) into Milli-Q water. The four histones were reconstituted into octamers and separated using a HiLoad 16/60 Superdex 200 column in buffer (2 M NaCl, 10 mM Tris, pH 7.5, 1 mM EDTA, pH 8.0, 5 mM b-mercapto-ethanol, and 0.2 mM PMSF). Equimolar fractions of the histones were pooled, concentrated on Amicon® Ultra Centrifugal Filter 3 kDa MWCO, mixed with 50% glycerol (v/v), and stored at −20 °C.

### Preparation of octamers and antibody-bead complexes

10 μg of recombinant octamers (2 μg/μl) were transferred to a 1.5 ml tube. These were cross-linked by the addition of 5 μl of octamer buffer (2 M NaCl, 10 mM Tris-HCl, pH 7.5, 1 mM EDTA, pH 8.0) and 0.27 μl of octamer cross-linking buffer (3.7% w/v formaldehyde in dH₂O), resuspended by pipetting. Following a 20 min. incubation at room temperature, the fixative was quenched for 5 min. at room temperature using 0.72 μl of 200 mM glycine in PBS solution (Sigma-Aldrich, 67419-1ML-F), and stored at 4 °C. The octamers were used within three months (concentration was 0.9 μg/μl). Each picoChIP reaction was supplemented with 2.44 μg of the cross-linked octamers. Protein A Dynabeads (Invitrogen, 10002D) were incubated with antibodies a day prior to chromatin sonication. The beads were thoroughly vortexed and the volume specified in Supplementary Data 5 was used for each picoChIP reaction and washed using RIPA buffer (10 mM Tris-HCl pH 8.0, 175 mM NaCl, 1 mM EDTA, 0.625 mM EGTA, 1.25% v/v Triton X-100, 0.125% w/v sodium deoxycholate, 1 x Halt-protease inhibitor cocktail (ThermoFisher, 78440), 1 mM PMSF, 20 mM sodium butyrate) two times. To improve handling, an excess of beads was washed using a PCR-strip magnetic rack (Diagenode), a 0.6 ml tube magnetic rack (manufactured in-house), and a hand-held strong neodymium magnet for complete separation when needed. The master mix of washed beads was prepared in a suitable tube (1.5 ml / 0.6 ml / 0.2 ml, Axygen), taking into consideration the intended number of ChIP reactions and the volumes given in Supplementary Data 5, followed by dilution in 20–100 μl RIPA per reaction. The corresponding volume of antibody

(Supplementary Data 5), followed by incubation on a 'head-over-tail' rotator (10 rpm) at 4 °C overnight (ON).

### Chromatin preparation

On the following day, transfer of the cross-linked samples was done using dry ice and thawing in a cold block for a few minutes, after which 120 μl lysis buffer (50 mM Tris−HCl pH 8.0, 10 mM EDTA pH 8.0, 0.8% w/v SDS, 1 x protease inhibitor cocktail, 1 mM PMSF, and 20 mM sodium butyrate) were added. Following 5-15 min. incubation of the samples in a cold block of approximately 6–8 °C, 30 μl of PBS supplemented with 20 mM sodium butyrate were added. Then, the samples were sonicated as indicated in Supplementary Data 6, using a Hielscher UP100H sonicator equipped with a 2 mm probe, configured at 27% amplitude and 0.5 s pulse intervals. Directly after sonication, the tubes containing 160 μl of sonicated chromatin were diluted with the appropriate volume of RIPA dilution buffer (10 mM Tris-HCl pH 8.0, 175 mM NaCl, 1 mM EDTA, 0.625 mM EGTA, 1.25% v/v Triton X-100, 0.125% w/v sodium deoxycholate, 1 x protease inhibitor cocktail, 1 mM PMSF, and 20 mM sodium butyrate) and processed further as indicated in Supplementary Data 7. Centrifugations were performed at 16,000 x $g$ for 10 min, at 4 °C. At this point, the 5% input samples were acquired from the 430 μl chromatin by storing 21.5 μl at 4 °C. Chromatin preparations intended for multiple ChIP reactions were divided into fractions of equal volume, containing the desired cell number chromatin equivalents, and the volume was adjusted to a final of 430 μl using volume adjustment buffer (40 parts PBS with 20 mM sodium butyrate, 120 parts lysis buffer, 270 parts RIPA dilution buffer).

### Washes and elution

Following the extensive incubation, the tubes were vortexed thoroughly, any droplets collected by a short centrifugation, the beads pelleted with a hand-held magnet, the supernatant removed, and the beads were transferred to 0.2 ml strip tubes using 150 μl of the first wash buffer. The washing steps for H2A.Z picoChIP-seq are given in Supplementary Data 8. The employed buffers provided wash steps of different stringency due to their compositions: RIPA-standard (10 mM Tris-HCl pH 8.0, 140 mM NaCl, 1 mM EDTA, 0.5 mM EGTA, 1% v/v Triton X-100, 0.1% w/v SDS, 0.1% sodium deoxycholate, 1 x protease inhibitor cocktail, 1 mM PMSF, 20 mM sodium butyrate) and RIPA-high (10 mM Tris-HCl pH 8.0, 300 mM NaCl, 1 mM EDTA, 0.5 mM EGTA, 1% v/v Triton X-100, 0.22% w/v SDS, 0.1% v/v sodium deoxycholate, 1 x protease inhibitor cocktail, 1 mM PMSF, 20 mM sodium butyrate). A magnetic rack was used to perform the washes. For each washing step, stringency was increased mechanically via vortexing for 3 × 5 s. on the highest setting, repeated twice, with 10 s. incubation on ice in between. Following the last washing step, TE buffer (10 mM Tris−HCl pH 8.0, 1 mM EDTA pH 8.0) was used to transfer each picoChIP sample to a new 0.2 ml tube, and the beads were eluted using 150 μl of elution buffer (20 mM Tris−HCl pH 7.5, 50 mM NaCl, 5 mM EDTA pH 8.0, 1% w/v SDS and 0.2 mg/ml RNase A). At this point, the 5% input chromatin tubes were mixed with 128.5 μl elution buffer and treated as the other samples going forward. RNase incubation was performed for 1 h at 37 °C with vigorous agitation (1250 rpm), followed by the addition of 1 μl of Proteinase K (20 mg/ml, NEB, P8107S) and incubation for 4 h at 68 °C with vigorous agitation.

### DNA purification, library preparation and quantification

The picoChIP and input samples were transferred to 1.5 ml tubes. SDS/RNase-free elution buffer (20 mM Tris−HCl pH 7.5, 50 mM NaCl, 5 mM EDTA pH 8.0) was added to a final volume of 415 μl, and the aqueous phase was sequentially extracted using equal volumes of phenol:chloroform:isoamylalcohol (Invitrogen, 15593-031) and chloroform:isoamylalcohol (Sigma-Aldrich, C0549). To the 400 μl of the last aqueous phase, 1 ml ice-cold ethanol, as well as 11 μl linear acrylamide (Thermo Fisher Scientific, AM9520) and 44 μl 1 M sodium acetate

(Invitrogen, AM9740) were added to precipitate the DNA at −80 °C at least overnight. For reconstitution, the pellets were centrifuged (swing-out buckets, 16,000 x g, 4 °C), and the DNA recovered using 15 μl of Qiagen EB (10 mM Tris, Qiagen, 19086) overnight at 4 °C. Illumina sequencing libraries were prepared using the Qiaseq ultralow input kit (Qiagen, 180492 + 180310), optimized to minimize material losses. These included prolonged incubations during the recoveries from AMPure XP beads (Beckman Colter, A63881), with >15 min. bead/sample incubation, and >15 min. DNA elution, as well as an increased volume of beads (1.2 volume equivalents) during the second bead extraction. Double-stranded DNA concentration was determined using the Qubit HS DNA kit (ThermoFisher Scientific, Q32851), while DNA fragment size analyses were performed using the TapeStation D1000 HS kit (Agilent, 5067–5585). Equal moles of each library were pooled, and sequential AMPure XP extractions (1.2 volume equivalents) reduced adapter dimers from the pool prior to sequencing. P7, P10 (CD1/Swiss), P12, NSN, SN, MII, 8-cell, morula and blastocyst samples were sequenced on a NovaSeq 6000 SP, paired end 50 bp, while P10 (C57BL6/N), zygote, 2-cell and 4-cell samples were sequenced on a NovaSeq X, paired end 150 bp at the Norwegian Sequencing Center (Ullevål, Oslo).

## Immunofluorescence and DNA staining

Immunostaining for CD1 mouse P7, P10, and P12 oocytes was performed in a 96-well plate, adjusting a previously reported protocol[66]. Briefly, embryos were fixed in 4% w/v paraformaldehyde for 15 min. Embryos were permeabilized in 0.3% w/v BSA, 0.1% v/v Triton X-100 in DPBS solution. Blocking was carried out in 0.3% w/v BSA, 0.01% v/v Tween-20 in DPBS. Embryos were incubated in blocking solution with 1:1000 α-H2A.Z antibody (Supplementary Data 5) for 1 h at room temperature. After three washes in blocking buffer, embryos were incubated with α-rabbit Alexa Fluor 594 (Invitrogen, A11037, lot: 2841610) at 1:1000 dilution for 1 h at room temperature, washed in blocking buffer three times, incubated in a 1:1000 solution of Hoechst 33342 (Thermo Fisher, 62249), washed twice and placed on a slide in SlowFade Gold (Invitrogen, S36936). Quantitative measurements of α-H2A.Z were obtained from stained mouse oocytes imaged on the same day using a Zeiss LSM 980 Airyscan 2 confocal microscope (Carl Zeiss MicroImaging GmbH, Jena, Germany) equipped with a Zeiss Plan-Apochromat 40x/0.94 air objective at the Advanced Light Microscopy Core Facility, Montebello node, Institute for Cancer Research, Oslo. Image acquisition was performed with ZEN 3.10, and the signal was quantified with NIS Elements AR (5.42.06 · Build 1821 · LO, 64-bit, Nikon) in batch mode. Briefly, the blue channel (Hoechst) was used for nucleus segmentation, a uniform background was subtracted from the red channel (AF594), and the sum of signal intensity within the nucleus was reported. For the statistical analysis, a preliminary Kruskal-Wallis test was done, followed by a post hoc Dunn test with Benjamini-Hochberg correction. Calculations were done in R[67] using rstatix[68] and visualized using ggplot2[69].

## Statistics & reproducibility

No statistical method was used to predetermine sample size. No data were excluded from the analyses. The experiments were not randomized. The Investigators were not blinded to allocation during experiments and outcome assessment.

## Data analysis

**Data sources.** All new ChIP-seq data are deposited at NCBI's Gene Expression Omnibus[70] under the accession number GSE293415. Datasets of H3K4me3, H3K27me3, H3K27ac, DNA methylation and ATAC-seq in mouse oocyte and early embryo development were sourced from[35–39,71,72] and downloaded from NCBI GEO[70] series GSE72784, GSE56879, GSE73952, GSE217970, GSM2041068, GSM6731319, GSM6731320, GSM3941360, GSM3941361, GSM5888085,

GSM5888086, GSM5888096 and GSM5888097. RNA seq data[31] were downloaded from ArrayExpress[73] (Accession Number P-MTAB-41287), and GEO[70] GSM1845293. Aggregated single-cell Repli-seq data of the maternal and paternal genomes in 2-cell mouse embryos were processed in[25] and obtained from Supplementary Data 7 in that paper. Additional 2, 4, 8, and 16-cell data were published in[42] and were downloaded from the GEO database[70] series GSE218365. The mean Replication Status (RS) values from all single-cells were used for analysis and visualization here. Bed-files with allele-specific coordinates for LADs in maternal and paternal 2-cell embryos[22] were downloaded from the GEO database[70] series GSE112551. Refseq gene annotations for mm10 were downloaded from the UCSC table browser[74,75], and coordinates for unique TSSs were defined based on this ($n = 32,166$). CGIs as defined in the mouse mm10 genome were also downloaded from the UCSC table browser[75]. CpG and GC densities from the mm10 reference genome sequences were downloaded from the UCSC genome browser[75,76] at https://hgdownload.soe.ucsc.edu/goldenPath/. A general mouse enhancer set titled ENC + EPD enhancers ($n = 37,473$) was sourced from the UCSC table browser[45,75,77], and a putative mouse oocyte enhancer set was sourced from[39]. A dataset of different embryonic gene categories in the mm10 genome was sourced from DBTMEE v2 transcriptome categories[78]. A dataset of LTR-initiated transcription units (LITs) ($n = 3384$) was obtained from[48]. Repeat sequences from the mouse genome was downloaded from the UCSC table browser[75] (Repeatmasker 21-10-2024[46]).

**ChIP-seq and RNA-seq data processing.** ChIP-seq and RNA-seq data were processed using an in-house Nextflow pipeline (https://github.com/lerdruplab/ew-qctrimalign) written according to nf-core guidelines[79] utilizing reproducible software environments from Bioconda[80] and Biocontainers[81]. Briefly, the pipeline was executed using Nextflow v23.4.1[82]. Single-end fastq files from P10 (C57BL6/N), zygote, 2-cell and 4-cell samples were hard-trimmed to 51 bp to match the read length of the remaining samples using Trim Galore v0.6.7 (https://github.com/FelixKrueger/TrimGalore) with the -hardtrim5 51 parameter. Quality trimming was performed using Trim Galore v0.6.7 with the parameter -novaseq 20. Trimmed reads were then mapped to mm10 using bowtie v1.3.0[83] with the parameter -m 1. Obtained bam files were sorted using samtools v1.2[84] and converted to bed format using bedtools v2.31.1[85]. Bed files were then used for downstream analysis in EaSeq[86,87].

**General data visualization.** Unless stated, values are FPKM normalized. Simple plots, including doughnut plots, scatter plots, line plots and tiles were generated in Microsoft Excel or using R[67]. Bubble plots were made in R[67] using tidyverse[88] and ggplot2[69]. The $\log_2$fold difference was calculated based on the difference between the observed and expected number for each analyzed feature. Statistical significance was assessed using a two-sided chi-square test with one degree of freedom for each comparison. The resulting p-values were adjusted for multiple testing using the Benjamini-Hochberg procedure to control the false discovery rate (FDR). The underlying data for the plots can be found in Supplementary Data 9. For correlation matrices, replicates from external studies were pooled. The genome was segmented into 10 kbp bins, and the FPKM values for each bin were used to compare the H2A.Z enrichment using Spearman's rank correlation coefficient. The matrices were generated in R using the corrplot package[89]. The following tools in Easeq[86,87] were used for data visualization and adjusted as described in figure legends: Genome browser tracks were generated using the "FillTrack" tool, Metagene plots were generated using the "Average" tool followed by the "Overlay" tool, Colored 2D-histograms were created with the "ZScatter" tool, and heatmaps and simple heatmaps were generated using the "HeatMap" and "Parmap" tools, respectively. Overlap between features was calculated using the "Coloc" tool with the setting "Distances measured from border to

border of the regions". For visualization of features throughout the mouse genome, the "Modify" tool was used to subdivide ("Homogenize") a region set listing sizes for each chromosome in the mm10 genome into 1 kbp bins ($n = 2,633,787$) and 10 kbp bins ($n = 263,389$), respectively.

**Clustering.** Regions were clustered in Easeq[87] using the "Cluster" tool set to perform k-means clustering using a k-value of 10. Initial k-means were assigned using the $k^{++}$ approach[90]. Clustering start point, end point, and offsets for each analysis are stated at the respective analysis and figure. For clustering across developmental stages, the zygote to 4-cell stages were not included due to relatively lower library complexities and, therefore, potential contributions of stochastic variation.

**H2A.Z peak calling and aggregation.** Peaks were called from samples of all developmental stages, except for the zygote to 4-cell stages, which had relatively lower library complexities and yielded few additional peaks. For P10 oocytes, reads from the two CD1 biological replicates were first pooled. Peaks from individual samples were called against corresponding input samples and then merged into one peak set ($n = 111,448$, Supplementary Data 2) in Easeq with default parameter settings as described in[87] using "Regionsets/Modify/Merge" in a consecutive manner. We then quantified all samples at the center of the peak ±1kbp using the "Quantify" tool in EaSeq. For clustering, quantified values based on H2A.Z from all developmental stages in the aggregate peak set were quantile normalized to ensure that H2A.Z signal from all samples contributed equally prior to clustering using the "Normaliz." and "ClusterP" tools in EaSeq, respectively. The principal component analysis was based on quantified and quantile-normalized H2A.Z enrichment in a 2 kbp window around the center of peaks. The analysis was carried out in R[67] using r-packages tidyverse[88] and plot3D[91]. To visualize enrichment/depletion of CpG density, H3K4me3 and H3K27me3 histone marks[35,36,71] in peaks compared to the genome-wide signal, Z-scores were calculated in the complete clustered peak set as well as the non-TSS/CGI peak set using the "Quantify" tool in Easeq. When replicates were available, average Z-scores were calculated prior to visualization.

**H2A.Z peak annotation.** Peak annotation was performed using ChIP-seeker v1.38.0[92] in R v4.3.3. Gene ontology analysis of H2A.Z peak clusters was performed in R v4.3.3 using gprofiler2[93] and visualized as a heatmap using ComplexHeatmap[94]. We used the full-stack ChromHMM segmentation data for the mm10 mouse assembly (github.com/ernstlab/mouse_fullStack_annotations)[30] to examine the overlap of the annotation with the complete H2A.Z peak set and the non-TSS/CGI peak set. Using shuffle bed from BEDtools[85], five randomized control regions of equal number and length to the peaks were created, and the overlaps were counted with annotate bed also from BEDtools. We depicted the average relative enrichment with the standard deviation as error bars. Additionally, gene assignments for the peaks were done by ChIP-Enrich[95] (chip-enrich.med.umich.edu) and the statistically significant GO:BP gene sets (FDR ≤ 0.05) were processed by Revigo[96] (revigo.irb.hr). Cytoscape was used to visualize the resulting networks[97] (cytoscape.org).

**H2A.Z at TSSs and CGIs.** For subgrouping, the H2A.Z signal at TSSs at the individual stages was quantified, and FPKM normalized at the center of the peak ±1 kbp using the "Quantify" tool and quantile normalized using the "Normaliz." tool, followed by clustering using the "ClusterP" tool in EaSeq. Zygote to 4-cell stages were not included due to relatively lower library complexities and, therefore, potential contributions of stochastic variation. RNA expression data obtained from[31], were quantified and FPKM normalized at TSSs (±500 bp), then quantile normalized using the "Normaliz." tool in EaSeq. H2A.Z signal and quantile-normalized RNA signal were then visualized as heatmaps

based on the clustering. To find enriched or depleted gene categories for each cluster, we calculated the number of genes from different embryonic transcriptome categories (DBTMEE v2 transcriptome categories[78]) in each cluster and visualized in Bubble plots as described above. To explore the relationship between H2A.Z, CGIs and other genomic features at TSSs, CGIs were colocalized to TSSs using the "Coloc." tool in EaSeq. Calculated distances were sorted based on proximity to the nearest CGI using the "Sort" tool. Correlations of H2A.Z, H3K4me3[35], and CpG densities both at TSSs ($n = 32,166$) and in the whole genome, divided into 1 kbp bins ($n = 2,633,787$), were calculated with Spearman's rank correlation coefficients in R[67].

**H2A.Z at repetitive elements.** To analyze the H2A.Z signal at repetitive elements, we colocalized H2A.Z peaks with repeat coordinates, and sub-selected repeats with a count of more than 200 occurrences in the aggregated peak set to ensure solid statistical analyses. To find enriched or depleted repeat categories for each cluster, repeat coverage as well as abundance was calculated for each peak cluster in the non-TSS/CGI peak set and used for visualization. Selected repetitive elements were chosen based on high abundance in clusters 1 and 2, and the P12 stage was used to visualize the H2A.Z signal at these. Repetitive elements were clustered at the start of the repeat ±3 kbp and sorted based on the mean H2A.Z signal in each cluster. Given the high H2A.Z signal at MTA RTs, LITs from[48] were sorted based on overlap to CGI and H2A.Z peaks, and used for visualization of local H2A.Z, H3K4me3 and H3K27me3 signals[35,36,71].

**Motif enrichment analysis and integration with CpG / G/C density data.** Identification of enriched DNA motifs were performed on aggregate H2A.Z peaks, which did not overlap with TSSs or CGIs, from maternal clusters (1, 2 and 3) and embryonic clusters (8,9 and 10). Sets of random control regions with matching distances to the nearest TSS were generated using the "Controls" tool in EaSeq, and sequences from the peaks and control regions were extracted using the "Get Sequences" tool in EaSeq (in the beta testing menu) and used for de novo motif enrichment analysis using the program MEME-ChIP[98]. Only hits with E-value ≤ $e^{-10}$ were considered. All genome sites of the resulting motifs were downloaded using the tool FIMO, imported as a dataset in Easeq and quantified ±1 kbp in the aggregated peak region set.

## Ethics declarations
All mouse experiments were approved and registered by the Norwegian Food Safety Authority (NFSA approved application/FOTS IDs: 7216, 10898, 24911, and 8743) and conducted in accordance with Norwegian regulation FOR-2015-06-18-761, which closely aligns with EU directive 2010/63/EU on stringent ethical and welfare standards to protect animals used for scientific purposes.

## Reporting summary
Further information on research design is available in the Nature Portfolio Reporting Summary linked to this article.

## Data availability
All ChIP-seq data generated for this study have been deposited at NCBI's Gene Expression Omnibus[70] under the accession number GSE293415. Publicly available datasets from the following sources were used: H3K4me3 (GSE72784 and GSE73952), DNAme (GSE56879), H3K27me3 (GSE73952 and GSE76687), H3K27ac (GSE217970), ATAC-seq (GSE217970, GSE134279 and GSE196520), RNA-seq (P-MTAB-41287 and GSE71434), Repli-seq (GSE237400 and GSE218365), and Lamin B1 (LADs) (GSE112551). Source data are provided with this paper.

## Code availability
Our Nextflow pipeline for ChIP-seq and RNA-seq data processing can be found at the GitHub repository https://github.com/lerdruplab/ew-

qctrimalign, while all other scripts are available at the GitHub repository https://github.com/lerdruplab/H2A.Z.

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

## Acknowledgments

We thank the members of the Lerdrup group, Dahl group, Reproductive Medicine Group, CRESCO, and the Center for Chromosome Stability for support and critical suggestions throughout this work. We are grateful to Thomas C.R. Miller for sharing computational infrastructure and Eva R. Hoffmann for support. In addition, we thank the Norwegian Sequencing Center, the Norwegian Transgene Center, and the staff at the animal facilities (Department of Comparative Medicine, Oslo University Hospital and Department of Comparative Medicine, Oslo University). We are also grateful to Ellen Skarpen and Vigdis Sørensen for assistance with imaging and image analysis at the HSØ Advanced Light Microscopy Core Facility, Montebello, and Gaustad nodes, Institute for Cancer Research, Oslo University Hospital. This work was supported by the Danish National Research Foundation DNRF115 (M.L.), the Novo Nordisk Foundation, NNF22OC0080710 (M.L.), NNF22OC0074308 (J.A.H.), Norwegian Centers of Excellence scheme grant 332713 - CRESCO (M.L. and J.A.D.), The South-Eastern Norway Regional Health Authority including 2023098 (J.A.D., M.F.), 2022047 (J.A.D., M.C.), grant 262652 - CanCell (R.E.), Research Council of Norway, 262484 (R.E.), 275286 (M.F.), The Carlsberg Foundation Equipment Grants (CF22-1209, CF21-0571), and Læge Sophus Carl Emil Friis og hustru Olga Doris Friis Legat (J.A.H.).

## Author contributions

M.F., M.L., and J.A.D. conceived the study. M.F., with input from J.A.D., designed the experiments. M.F. and M.C. carried out picoChIP-seq experiments. R.E. carried out the generation of recombinant octamers used in picoChIP-seq. T.S., I.J., R.S. and S.K. under supervision by G.G., K.T.D., P.F., M.B., and J.A.D. performed mouse work and ovary collection. M.F., T.S., M.C., J.A.H., M.I., A.M., A.G.S., M.V-R., G.G., and J.A.D. planned and performed mouse oocyte and embryo collections, and E.K. and A.K. contributed supervision of A.G.S. and A.M. T.S. performed immunofluorescence and imaging. M.C. performed image quantification under the supervision of T.S. E.I., M.S., M.C., and M.Z. with input, assistance and supervision from M.L. performed the data analysis. M.F., E.I., M.Z., T.S., M.C., M.S., M.L., and J.A.D. discussed the data and provided critical input. M.Z., E.I., and M.L. with assistance from M.F., P.F., T.S., M.S., and J.A.D. prepared the manuscript. All authors read and commented on the manuscript.

## Competing interests

The authors declare no competing interests.

## Additional information

¹Department of Microbiology, Oslo University Hospital, 0372 Oslo, Norway. ²CRESCO, Centre for Embryology and Healthy Development, University of Oslo, 0373 Oslo, Norway. ³DNRF Center for Chromosome Stability, Department of Cellular and Molecular Medicine, University of Copenhagen, DK-2200 Copenhagen, Denmark University of Copenhagen, DK-2200, Copenhagen, Denmark. ⁴Department of Reproductive Medicine, Oslo University Hospital, 0855 Oslo, Norway. ⁵Institute of Clinical Medicine, University of Oslo, 0372 Oslo, Norway. ⁶Norwegian Transgenic Center (NTS), University of Oslo, 0373 Oslo, Norway. ⁷CRESCO, Centre for Embryology and Healthy Development, Department of Biotechnology, University of Inland Norway, Hamar, Norway. ⁸Department of Molecular Medicine, Institute of Basic Medical Sciences, University of Oslo, Oslo, Norway. ⁹Centre for Cancer Cell Reprogramming, Institute of Clinical Medicine, Faculty of Medicine, University of Oslo, Oslo, Norway. ¹⁰These authors contributed equally: Madeleine Fosslie, Erkut Ilaslan. ✉e-mail: mikaza@sund.ku.dk; j.a.dahl@medisin.uio.no; mlerdrup@sund.ku.dk

