## [Transparent Peer Review file · Nature Communications]

Major waves of H2A.Z incorporation during mouse oogenesis and preimplantation embryo development

Corresponding Author: Dr Mads Lerdrup

Version 0:

Reviewer comments:

Reviewer #1

(Remarks to the Author)

This study presents a rigorous experimental analysis of H2A.Z genome-wide deposition patterns during different stages of mouse oogenesis. The authors explore various factors that influence the deposition patterns, including genomic regions, DNA sequence motifs, sequence composition and histone post-translational modifications H3K4me3 and H3K27me3.

Overall, the study appears to be well-executed, though I am not very familiar with all the experimental techniques employed. However, the manuscript is quite difficult to follow due to the numerous correlations and interdependencies between various factors. To improve clarity, I recommend including a summary figure that visually integrates the key findings, and a more focused interpretation of the results would be also beneficial.

A major point that is currently missing is molecular-level interpretation, particularly in the context of H2A.Z-containing nucleosomes. Previous studies have shown that H2A.Z nucleosomes are less thermodynamically stable and more prone to DNA unwrapping from the histone core (PMIDs: 34643712, 36765119). This could lead to a greater solvent-accessible surface exposure and could explain the observed correlation between H2A.Z deposition and active gene expression, as increased accessibility may facilitate transcriptional machinery binding and/or lower transcriptional barrier. Additionally, the co-deposition of H2A.Z with H3.3, a variant also associated with chromatin instability, could be discussed (PMID: 17575053).

Another important aspect concerns sequence composition and motifs. The authors report co-occurrence between sequence repeats and H2A.Z deposition, yet many repeat elements are typically excluded from nucleosome regions. The manuscript lacks a clear connection between the observed sequence features and nucleosome positioning: do the reported correlations reflect nucleosome enrichment or depletion in these regions?

Some other minor comments:

- Figures 2 and 3 report very low effect sizes ($\log_2(\text{obs}/\text{exp})$) of less than 1, I am not sure if it is worth reporting them in the main text or Supplementary Materials.
- Figure 2 does not have an explanation of panel e).

(Remarks on code availability)

The authors use the standard codes for peak calling.

Reviewer #2

(Remarks to the Author)

This manuscript by Fossli and Ilasslan and colleagues describes the genome-wide localization of the histone variant H2A.Z during mouse oogenesis and in early embryos. Suitable controls are provided to demonstrate reliability of the data, as far as can be demonstrated. This manuscript thus provides a useful resource to the community, with the first demonstration of H2A.Z genomic binding profiles in growing mouse oocytes. They present interesting and important findings from these profiles, including a major wave of H2A.Z incorporation in growing oocytes between P7 and P12, both at CGI TSS and non-TSS regions, and associations between H2A.Z accumulation in oocytes and epigenetic features in fertilized embryos, such

as histone marks, lamina-associated domains and replication timing. An analysis of the association between H2A.Z genomic localization and distinct families of TEs in oocytes and early embryos is also presented, revealing intriguing stage-specific associations. The data is very well-presented.

However, I have concerns over the interpretation of the data. The title of the manuscript refers exclusively to oogenesis, but the manuscript often describes H2A.Z incorporation and its association with epigenetic processes in 'early embryos'. However, only late-stage preimplantation embryos are included (8-cell to blastocyst) and used to represent 'early embryos' in general. Data from early-stage preimplantation embryos is not included in their analysis (i.e. zygotes to 4-cell stage). I understand that such data is previously published and analysed in PMID: 3571767 and the authors wish to avoid redundant analysis. However, this significantly impacts the interpretations of the data and their conclusions can be misleading as they frequently make generalizations about early embryos. I describe several examples of such concerns below.

1. The use of the term 'constant' throughout the manuscript for H2A.Z regions that are enriched in H2A.Z across all stages analysed is misleading as it is not clear whether these regions retain H2A.Z during the intermediate stages not studied. The assumption and likely overestimation of constant H2A.Z regions pervades the manuscript, for example in the abstract 'The existence of changes in H2A.Z distribution that persist across related developmental stages enable preservation of epigenetic information despite major concurrent changes in H3K4me3, H3K27me3, and DNA methylation.' A similar concept is put forward in the discussion (lines 388-396). I find this interpretation too strong as the dynamics of H2A.Z in early embryos are likely to be significantly underappreciated due to the lack of data. While it could be argued that one could increase the time resolution *ab infinitum* to demonstrate constant binding, I think profiling should be performed at each cell cycle, during this period of high epigenetic dynamics to demonstrate constant enrichment and inheritance. Likewise, the term 'oocyte-specific' for clusters 1-3 in figure 3a is not necessarily accurate, due to the missing stages.

2. In the analysis of H2A.Z accumulation at TSSs in Figure 2a, they conclude that only clusters VII and VIII are oocyte or embryo-specific. These conclusions cannot be generalized as such. A significant accumulation of H2A.Z peaks at promoters is shown at the 2-cell stage in PMID: 35717671, for example, which is missed by this manuscript.

3. The authors analyse processes that occur during the 'missing' stages of development, including zygotic genome activation (zygote to 2-cell stage), LAD formation (zygote) and dynamics (2-cell stage) and replication timing establishment (4-cell stage). While their focus is on correlation between the incorporation of H2A.Z in oocytes and later epigenetic processes, the lack of knowledge of whether or not H2A.Z is inherited or lost at these sites limits the interpretation of these data and leads to misleading conclusions. As one example, they show that clusters 2 and 3 are enriched in genes associated with major ZGA (Figures 2a-c), but we don't know if they remain associated with H2A.Z at the time of their activation in the 2-cell stage. They conclude that: (lines 178-179) 'Overall, H2A.Z was located at genes expressed at ZGA and MGA both during oogenesis and preimplantation embryo development.' This is misleading as they are unable to determine whether H2A.Z is located at ZGA genes in the relevant stage of preimplantation development when they are expressed. This leaves a significant gap in the insight put forward by this manuscript.

4. They find an association between oocyte-specific H2A.Z peaks in cluster 3 (Figure 3a) and paternal LADs. They state that oocyte-specific H2A.Z peaks are maternally inherited (line 248) and thus counteract maternal LAD establishment. However, I don't see the evidence for inheritance as data in zygotes is not shown. Figure 3a shows that cluster 3 is no longer enriched in H2A.Z in 8-cell stage embryos. In any case oocyte-specific peaks could not be maternally-inherited by definition, as they would not be oocyte-specific if they were also present in embryos. Overall, the enrichment of H2A.Z within LADs/iLADs in the embryo is not shown, which limits the interpretation of this data.

5. An analysis of association between H2A.Z dynamics and H3K4me3/H3K27me3 is shown (Figure 3b), considering the known association between these epigenetic features and the atypical distributions of these marks in early embryos. However, the insight gained from this analysis is limited as the stages during which dramatic changes in distribution of these marks occurs (zygote to 2-cell) is missing.

Overall, if the authors wish to make conclusion concerning general patterns of H2A.Z accumulation in oocytes and early embryos and association between H2A.Z incorporation and epigenetic dynamics in early embryos, data of H2A.Z incorporation in early embryos (zygote to 4-cell) should be included to provide a more comprehensive picture of H2A.Z dynamics during these stages of development.

Other major points:

1. Replication timing

When discussing replication timing the authors fail to cite two major papers characterising replication timing during preimplantation development: PMID: 35256805 and 39198647 (lines 253-256). These papers describe the appearance of the replication timing program from the 4-cell stage, in contrast to the 2-cell stage they state. The text should be altered to reflect this. In addition, analysis of replication timing at the 4-cell and 8-cell stage should be conducted in relation to H2A.Z incorporation in oocytes to determine whether the tendency observed in 2-cell stage (Figure 3d) is robust.

2. Enhancer analysis.

The authors used a generic list of enhancers from USCS for their analysis. However, many distinct oocyte and embryo-specific enhancers are also proposed to exist (PMID: 38839978). Is H2A.Z enriched in these at the corresponding stages? Is H2A.Z correlated with the presence of H3K27ac at enhancers at the respective stages?

3. Gene expression analysis

How do patterns of H2A.Z incorporation during oogenesis correlate with patterns of RNA expression during oogenesis prior to transcriptional shutdown?

4. TE expression analysis.

The stage-specific association between H2A.Z localization at distinct TE families is intriguing. Many TEs are expressed and act as alternative promoters in these stages of development. A broader analysis of the association between H2A.Z accumulation and TE expression should be conducted. In general, does H2A.Z enrichment correlate or anticorrelate with expression of TEs? Are TE families enriched with H2A.Z at different stages more likely to be expressed or repressed than those without? Does LIT expression correlate with H2A.Z accumulation? Is H2A.Z more or less enriched at insertions more highly expressed and acting as alternative promoters than those that are silent?

5. Embryo collection

In the methods they describe culture of embryos to E3.5 at which point 8-cell, morula and blastocysts were collected. If this is the case, then the data from 8-cell and morula embryos are not reliable as these embryos are delayed in development and are thus abnormal. 8-cell and morula embryos should be collected at around 70 and 80 h phCG respectively. Please also add the embryo growth protocol to the GEO accessions.

(Remarks on code availability)

The authors have made the code available at the above address. However, I am not proficient in coding so I am unable to assess this.

Reviewer #3

(Remarks to the Author)

This study centers on the profiling and analyses of H2A.Z dynamics during oogenesis and preimplantation development. Previous H2A.Z study (PMID 35717671) mainly focused on preimplantation embryos starting from MII eggs to blastocysts. The current study fills the knowledge gap by evaluating H2A.Z dynamics in early growing oocytes including postnatal day 7 (P7), P10, P12, and fully grown oocytes. The study made a few interesting observations including 1) dramatic increase of H2A.Z signal starting from P12 oocytes; 2) strong correlation between H2A.Z and CGI transcription start sites; 3) associations between H2A.Z enrichment and replication timing and lamina associated domains; 4) enrichment of repetitive elements in H2A.Z peaks.

Although this study provides some new insights, I found it largely descriptive and correlative with little functional characterizations. Previous study (PMID 35717671) has 1) established the H2A.Z landscape in preimplantation embryos, 2) demonstrated that H2A.Z is critical for lineage commitment, and 3) characterized how chromatin remodelers may regulate H2A.Z deposition in early development. The current study appears to have a moderate conceptual advance to the field unless functional demonstration of critical roles of H2A.Z in oogenesis.

Following are some specific comments for authors to consider addressing to further improve the story.

- 1) Although the bioinformatic analyses describing potential interactions/associations between H2A.Z, CGI, RNA, H3K4me3, H3K27me3, LAD, replication timing, and retrotransposons are interesting, functional characterization is essential to establish causal relationships.
- 2) Immunostaining experiments are recommended to further verify the global increase of H2A.Z from P7/P10 to P12 oocytes.
- 3) How about H2A.Z expression levels in oogenesis? Whether H2A.Z expression changes may explain the H2A.Z dynamics revealed by low-input ChIP-seq?
- 4) The Fig. 2b RNA FPKM max is quite low (i.e., 3), which may exclude many genes that express higher than this cutoff. As a result, this heatmap could be misleading as it does not show clear patterns of maternal and/or zygotic genes.
- 5) The comparison of LAD vs. H2A.Z in Fig3 is confusing. The LAD data were from 2-cell stage whereas none of the clusters in Fig. 3a-b include 2-cell stage.

Minor comments:

- 1) Fig. s1c is not clear to readers. What's the interpretation? Whether higher Z-score is better?
- 2) The number of biological replicates is unclear. Table1 suggests that only 1 replicate was generated for each stage.

(Remarks on code availability)

Version 1:

Reviewer comments:

Reviewer #1

(Remarks to the Author)

The authors have addressed my comments and overall, the paper has significantly improved. However, I have one comment about using ATAC-seq for nucleosome positioning. ATAC-seq has many limitations and it provides rather short fragments (much less than 147 base pairs) and usually ATAC-seq peaks are associated with nucleosome-free regions. So the authors should be very careful in interpreting their ATAC-seq results.

Some wording is still vague. Just one of many examples: “H2A.Z peaks are correlated to H3K4me3” – correlation coefficients can be calculated between the numerical values, not between peaks and modifications. “H2A.Z is strongly 451 correlated to paternal LADs”...

“H2A.Z 430 differs markedly from previously assessed epigenetic marks” – H2A.Z is not a modification mark.

(Remarks on code availability)

I did not run the code. I noticed that the README file is almost empty and contains very little information.

Reviewer #2

(Remarks to the Author)

I appreciate the authors efforts in adding the new data for zygote to 4-cell stage embryos throughout the manuscript. With this and the additional changes the majority of my comments have been addressed. It appears indeed that the addition of these stages does not change the overall clustering in Figure 3a or the conclusions made regarding persistent, maternal and embryonic clusters. However, in my view it cannot be ruled out that the lack of changes to the overall dynamics is not due to the low library complexity and the inability to call peaks, as they mention. This consideration should be added to the manuscript, for example in the limitations of the study.

Indeed, Figure 3a excludes the data from zygote to 4-cell for the peak clustering (stated in lines 224-225; this should be made clear also in the Figure legend) and thus potential dynamics of these stages may be missed. For example, in 'Fig 3 with new clustering' in the file for the reviewer, cluster 8 now has signal appearing at the 2-cell stage for example. This is apparent with the limited number of called peaks from the new data and thus if more peaks could be called the pattern could change more dramatically.

Regarding the replication timing, the reference Nakatani et al (ref 40) should also be cited here (lines 282-284): ‘The timing program correlates with transcription, LADs, genome compartmentalization and inherited histone modifications with consistent differences between the parental genomes’.

(Remarks on code availability)

Reviewer #3

(Remarks to the Author)

In this revised manuscript, the authors generated new H2A.Z ChIP-seq data for zygotes, 2-cell, and 4-cell embryos, and integrated these data and analyses throughout the manuscript. This addition improves the quality of the study. However, I still have the following major concerns:

Major Comments

1. Data quality in early embryos

I appreciate the authors' effort to generate new data despite the challenges of collecting hundreds of embryos for low-input ChIP. However, the data quality at the zygote to 4-cell stages appears considerably lower than that of other developmental stages, with only ~2 million non-duplicated reads. Consequently, the authors were unable to call peaks from these stages (lines 153–156).

While these data may provide some insights when used in genome-wide analyses, they should be interpreted with caution. For example, the reported global reduction of H2A.Z at TSSs in 2-cell embryos (Fig. 1b and Fig. 2a) may reflect technical limitations rather than biological differences. I strongly recommend either generating higher-quality datasets for these stages or incorporating already published datasets, as suggested by Reviewer #2.

2. Lack of biological replicates

With the exception of P10 oocytes, most stages appear to have only one biological replicate. Although I understand the scarcity of oocytes and early embryos, reproducibility is essential to ensure confidence in the findings. The authors nicely validated their blastocyst data against published datasets (Fig. S1d,e). I suggest applying a similar validation strategy for other stages by comparing with recently published H2A.Z datasets in oocytes and embryos. This would provide cross-validation and strengthen confidence in the conclusions.

Minor Comment

- A summary table of sequencing metrics (e.g., total reads, mapped reads, deduplicated reads) should be included for all datasets.

(Remarks on code availability)

Version 2:

Reviewer comments:

Reviewer #1

(Remarks to the Author)

My comments have been addressed.

(Remarks on code availability)

the description of codes is still limited. A log list of scripts but it is not clear what they are for.

Reviewer #2

(Remarks to the Author)

My outstanding comments have now been addressed and I can recommend publication of this manuscript.

(Remarks on code availability)

Reviewer #3

(Remarks to the Author)

The authors have conducted extensive comparisons between their datasets and previously published datasets, as I had suggested. These analyses are informative and help guide readers in accurately interpreting results generated using different methodologies. For example, crosslinking appears to cause stronger enrichment of H2A.Z at transcription start sites (TSS) compared with native ChIP conditions (new Fig. S2c). This may also explain why Liu et al. detected relatively weak H2A.Z signals at the MII stage, which used a native ChIP condition.

I appreciate the authors for performing these additional analyses and for providing a comprehensive comparison with prior datasets generated using distinct experimental approaches. I have no further concerns and support publication of this work.

(Remarks on code availability)

POINT-BY-POINT RESPONSE TO REVIEWER COMMENTS

Color codes:

Reviewer comment

Authors' comments

Authors' actions

Reviewer #1 (Remarks to the Author):

This study presents a rigorous experimental analysis of H2A.Z genome-wide deposition patterns during different stages of mouse oogenesis. The authors explore various factors that influence the deposition patterns, including genomic regions, DNA sequence motifs, sequence composition and histone post-translational modifications H3K4me3 and H3K27me3.

Authors' comments: Thank you for taking the time to review our manuscript and for recognizing the extent of the experiments and analyses.

Your recommendations for further discussion of the stability of H2A.Z nucleosomes and additional interpretation of repeats, H2A.Z, and chromatin structure have strengthened our findings. Below, we have addressed each of the comments point-by-point and the specific changes made in response to the feedback.

Overall, the study appears to be well-executed, though I am not very familiar with all the experimental techniques employed. However, the manuscript is quite difficult to follow due to the numerous correlations and interdependencies between various factors. To improve clarity, I recommend including a summary figure that visually integrates the key findings, and a more focused interpretation of the results would be also beneficial.

Authors comments: Thank you for the kind guidance on how to improve clarity in the manuscript. We agree that the complexity of these data requires an extraordinary focus on the communication.

Authors actions: To improve clarity, we have restructured parts of the results and discussion sections and added a summary figure (revised manuscript, Fig. 6).

A major point that is currently missing is molecular-level interpretation, particularly in the context of H2A.Z-containing nucleosomes. Previous studies have shown that H2A.Z nucleosomes are less thermodynamically stable and more prone to DNA unwrapping from the histone core (PMIDs: 34643712, 36765119). This could lead to a greater solvent-accessible surface exposure and could explain the observed correlation between H2A.Z deposition and active gene expression, as increased accessibility may facilitate transcriptional machinery binding and/or lower transcriptional barrier. Additionally, the co-deposition of H2A.Z with H3.3, a variant also associated with chromatin instability, could be discussed (PMID: 17575053).

Authors comments: These are good points and we have now discussed these topics in the revised manuscript.

Authors actions: We have now added a section on H2A.Z stability, versatility, DNA-interaction and H3.3 and included the suggested references in the discussion (Lines 484-495).

Another important aspect concerns sequence composition and motifs. The authors report co-occurrence between sequence repeats and H2A.Z deposition, yet many repeat elements are typically excluded from nucleosome regions. The manuscript lacks a clear connection between the observed sequence features and nucleosome positioning: do the reported correlations reflect nucleosome enrichment or depletion in these regions?

Authors comments: We agree that the relationship between repeats, nucleosome density, and H2A.Z deserved further attention, and we have sought to address this through additional analyses integrating motif abundance and previously published ATAC-seq data. While the ATAC-seq data did not have the library complexity (6-13 million deduplicated mapped reads) to perform modelling of nucleosome positioning at individual loci, we have used these datasets to investigate the location and size of nucleosome free regions in the context of H2A.Z signal and repeat types.

Authors actions: We have now included chromatin accessibility data from ATAC-seq experiments published in three studies (PMIDs: 36864102, 31919188, 38839978) to investigate the chromatin structure at H2A.Z incorporation sites (Figs. 3b and S3c) and in the context of repeats (Figs. 5b+d and S6a+b). In general, we see a positive relationship between stage-specific H2A.Z and chromatin accessibility (see e.g. revised Fig. 3b) as well as a general link between chromatin accessibility at CGIs and H2A.Z density (revised Fig 5d). We also observed that H2A.Z occurred next to regions with high accessibility at MTA repeats (possibly nucleosome-free regions, see Fig. 5b, S6a). Finally, we have directly visualized the abundance of the motifs we identified as enriched at certain stage-specific H2A.Z peaks and find that they are remarkably specific for the clusters of stage-specific H2A.Z (Fig. S5c). While we performed similar analyses for the simple repeat types found as enriched at H2A.Z peaks in Fig. 4a, we did not identify a strong relationship to any stage-specific H2A.Z (See figure below).

Some other minor comments:

- *Figures 2 and 3 report very low effect sizes ($\log_2(\text{obs}/\text{exp})$) of less than 1, I am not sure if it is worth reporting them in the main text or Supplementary Materials.*

- *Figure 2 does not have an explanation of panel e).*

Authors comments: We have looked further into the underlying numbers. In figure 2c for the Minor ZGA genes in cluster 4 the observed and expected numbers are 468 and 299, respectively. Likewise, for MGA genes in cluster 3 the observed and expected numbers are 340 and 215, respectively. It can be argued that these effect sizes somewhat limit the statistical power, and if the reviewer maintains this view, then we will be happy to move the figure into the supplement. In figure 3c for the maternal genome only category for cluster 2, the observed and expected numbers are 2303 and 1521, respectively. In the paternal only category for cluster 3 the observed and expected numbers are 2518 and 1694, respectively. As these effect sizes are considerable and result in extremely low adjusted p-values, we therefore find that the relationships are highly unlikely to be affected by stochastic effects and limitations in the datasets.

Authors actions: We have added a supplementary table with effect sizes (Table S3) together with the remaining intermediate and final values for the plots.

R1_4: - Figure 2 does not have an explanation of panel e).

Authors comments: Thank you for making us aware of this error.

Authors actions: We have amended the missing figure text

Reviewer #1 (Remarks on code availability):

The authors use the standard codes for peak calling.

Authors comments:

Our code availability now states:

“Our Nextflow pipeline for ChIP-seq and RNA-seq data processing can be found at the GitHub repository <https://github.com/lerdruplab/ew-qctrimalign>, while all other scripts are available at GitHub repository <https://github.com/lerdruplab/H2A.Z>.”

Reviewer #2 (Remarks to the Author):

This manuscript by Fossli and Ilaslan and colleagues describes the genome-wide localization of the histone variant H2A.Z during mouse oogenesis and in early embryos. Suitable controls are provided to demonstrate reliability of the data, as far as can be demonstrated. This manuscript thus provides a useful resource to the community, with the first demonstration of H2A.Z genomic binding profiles in growing mouse oocytes. They present interesting and important findings from these profiles, including a major wave of H2A.Z incorporation in growing

oocytes between P7 and P12, both at CGI TSS and non-TSS regions, and associations between H2A.Z accumulation in oocytes and epigenetic features in fertilized embryos, such as histone marks, lamina-associated domains and replication timing. An analysis of the association between H2A.Z genomic localization and distinct families of TEs in oocytes and early embryos is also presented, revealing intriguing stage-specific associations. The data is very well-presented.

Authors' comments: We sincerely appreciate the thorough assessment of our manuscript, the recognition of the extent and importance, and are grateful for the constructive points raised for revision.

In particular, the suggestion below to generate additional data within the timespan from MII oocytes to 8-cell embryos was highly relevant, and we have conducted additional experiments with zygote, 2-cell, and 4-cell embryos to strengthen the experimental data as well as conclusiveness of our study.

However, I have concerns over the interpretation of the data. The title of the manuscript refers exclusively to oogenesis, but the manuscript often describes H2A.Z incorporation and its association with epigenetic processes in 'early embryos'. However, only late-stage preimplantation embryos are included (8-cell to blastocyst) and used to represent 'early embryos' in general. Data from early-stage preimplantation embryos is not included in their analysis (i.e. zygotes to 4-cell stage). I understand that such data is previously published and analysed in PMID: 3571767 and the authors wish to avoid redundant analysis. However, this significantly impacts the interpretations of the data and their conclusions can be misleading as they frequently make generalizations about early embryos. I describe several examples of such concerns below.

Authors comments: We fully agree with the point raised by the reviewer and realize that inclusion of these early embryo stages would strengthen our study and several of the claims. Rather than using published data, we decided to generate new H2A.Z ChIP-seq data from zygotes, 2-cell, and 4-cell embryos using our validated cross-linking ChIP-seq method (PMID: 27626377, 32231309) that we used to generate the data presented in the first submission. We found this preferable compared to comparisons based on H2A.Z data generated using native ChIP where limited enrichment was observable in oocytes and the earliest embryonic states in (Liu et al, PMID: 3571767).

Authors actions: We have harvested 200 zygotes, 208 cells from 2-cell embryos and 172 cells from 4-cell embryos (see table 1), and generated H2A.Z ChIP-seq data from these stages for our study. All relevant figures have been updated (Figs. 1a-c, e, 2a, e, S2a-c, 3a, S3c, 4d, S4a, S6a). To be transparent, we would like to mention that the new data did not achieve the same library complexity as the previous H2A.Z ChIP-seq samples in a timely manner. The quality of the data is sound, but the read depth is lower compared to the samples from the initial submission. This is not unusual for low cell number samples, and the data are still very much of value as long as one takes this into account when doing comparisons. The approximately 2 million uniquely mapped deduplicated reads we have obtained for these samples suffice for hypotheses testing, profiling of genome-wide enrichment, and reliably resolving general changes in enrichment. However, peak-calling is generally more challenging for these numbers of reads and as one would expect we obtained relatively few additional peaks. Moreover, increased stochastic variation from limited library complexity is likely contributing to increased noise in the

unsupervised clustering and therefore reduce the explained variance in principal component analysis, possibly explaining the reduction from 43.6% to 17.5% for PC1 and from 22.8% to 14.7% for PC2 (Fig. 1e). Accordingly, we tested whether peaks and enrichment from the new zygote, 2-cell, and 4-cell data influenced our clustering and any conclusions made, but the overall profiles, changes, and ‘waves’ in H2A.Z enrichment were largely similar to that seen with the original clustering (See figure below). The main changes in the cluster composition is in line with the extent usually seen when the stochastic k-means clustering is rerun on the same data. Based on all these factors, we therefore decided to keep the original aggregate peak-sets and clustering in the revised manuscript.

1. The use of the term ‘constant’ throughout the manuscript for H2A.Z regions that are enriched in H2A.Z across all stages analysed is misleading as it is not clear whether these regions retain H2A.Z during the intermediate stages not studied. The assumption and likely overestimation of constant H2A.Z regions pervades the manuscript, for example in the abstract ‘The existence of changes in H2A.Z distribution that persist across related developmental stages enable preservation of epigenetic information despite major concurrent changes in H3K4me3, H3K27me3, and DNA methylation.’ A similar concept is put forward in the discussion (lines 388-396). I find this interpretation too strong as the dynamics of H2A.Z in early embryos are likely to be significantly underappreciated due to the lack of data. While it could be argued that one could increase the time resolution *ab infinitum* to demonstrate constant binding, I think profiling should be performed at each cell cycle, during this period of high epigenetic dynamics to demonstrate constant enrichment and inheritance. Likewise, the term ‘oocyte-specific’ for clusters 1-3 in figure 3a is not necessarily accurate, due to the missing stages.

Authors comments: We agree that the use of ‘constant’ and ‘specific’ to describe the subpopulations of H2A.Z included some potential inferences that this type of data does not address. Besides the missing stages that the reviewer rightfully pointed out, it is likely that both changing and unchanging H2A.Z enrichment is the result of dynamic equilibriums with potentially high levels of ongoing incorporation and eviction. We appreciate the opportunity to

clarify and adjust the terminology to avoid unintentional connotations. In addition, this motivated the generation of new data to better resolve the missing stages.

Authors actions: We have clarified in Fig 3a that the categories refer to the enrichment and not to H2A.Z itself, and renamed the groups from ‘Constant’ to ‘Persistent’, from ‘Early / Late Oocyte specific’ to ‘Early / Late Maternal’, and from ‘Embryo specific’ to ‘Embryonic’. We have also changed phrases throughout the manuscript text to specify that these categories describe the enrichment. Finally, we changed phrasing in the abstract from “(...) We identify distinct patterns of H2A.Z signal at oocyte-specific, embryo-specific, and constant H2A.Z loci. (...) While constant H2A.Z is strongly associated with CpG islands (...)” to “(...) We identify distinct patterns of H2A.Z signal at *maternal loci*, *embryonic loci*, and *loci with persisting enrichment*. (...) While *persisting* H2A.Z enrichment is strongly associated with CpG islands”. We hope that the reviewer finds these statements to be more in line with the data and are open to additional specific requests, if the reviewer feels that there are residual implied relationships that are not fully supported by the data types and assays.

2. In the analysis of H2A.Z accumulation at TSSs in Figure 2a, they conclude that only clusters VII and VIII are oocyte or embryo-specific. These conclusions cannot be generalized as such. A significant accumulation of H2A.Z peaks at promoters is shown at the 2-cell stage in PMID: 35717671, for example, which is missed by this manuscript.

Authors comments: As described above, we have now changed the terminology to avoid the use of ‘specific’. While we are generally appreciative of the referred work (PMID: 35717671), the global absence of H2A.Z signal in oocytes and henceforth the increase in 2-cell promoter signal observed in this study, is possibly related to assay differences. While it is hard to pinpoint the exact reasons for these global differences observed earlier, two studies published in Nat. Struct. Mol. Biol. during the revision process (PMID: 40514539, 40514538) also identify notable localized H2A.Z enrichment in mouse oocytes. While these two new studies bring forth a strong set of genetic models to study interdependence between different histone marks, our work provides considerably more stages during oocyte and embryo development as well as a stronger investigation of relationships to repetitive elements, LADs, and replication timing.

Authors actions: In addition to updating terminology in text and Fig. 3a as stated above, we have also included references to the new Nat. Struct. Mol. Biol. studies and a section in discussion focused on differences and similarities between the studies.

3. The authors analyse processes that occur during the ‘missing’ stages of development, including zygotic genome activation (zygote to 2-cell stage), LAD formation (zygote) and dynamics (2-cell stage) and replication timing establishment (4-cell stage). While their focus is on correlation between the incorporation of H2A.Z in oocytes and later epigenetic processes, the lack of knowledge of whether or not H2A.Z is inherited or lost at these sites limits the interpretation of these data and leads to misleading conclusions. As one example, they show that clusters 2 and 3 are enriched in genes associated with major ZGA (Figures 2a-c), but we don’t know if they remain associated with H2A.Z at the time of their activation in the 2-cell stage. They conclude that: (lines 178-179) ‘Overall, H2A.Z was located at genes expressed at ZGA and MGA both during oogenesis and preimplantation embryo development.’ This is misleading as they are unable to determine whether H2A.Z is located at ZGA genes in the

relevant stage of preimplantation development when they are expressed. This leaves a significant gap in the insight put forward by this manuscript.

Authors comments: We appreciate the insightful views and agree that the inclusion of H2A.Z data from zygotes, 2-cell, and 4-cell embryos would fill out gaps and support the validity of such statements. The new H2A.Z ChIP-seq data added during revision do show that the enrichment is inherited in the zygote and at least partially in the 2-cell embryo. We don't want to push conclusions on the 2-cell stage as the drop in signal intensity is global, and in our experience this can potentially have multiple causes, including biologically driven loss of the target, enrichment variation and a global increase in signal at other loci which due to the compositional nature of the library data leads to a reduced read count at the studied loci (PMID: 29608657). The strong transcriptional induction of H2A.Z genes (H2AFV and H2AFZ) together with a higher basal incorporation of H2A.Z throughout the genome could lead to such an effect. However, to assess the reviewers' questions more directly, we have also visualized the H2A.Z enrichment at subsets of genes involved in Minor, Major ZGA, and MGA (See figure below), and despite the (globally reduced signal) there is also persisting enrichment at those TSSs across stages.

We would like to add that our interpretations did not imply a direct role, but mainly described the genome-wide relationships, and we have adjusted phrases to clarify this. We hope that the reviewer finds that we now present a balanced view supported by the data.

Authors actions: We have updated Fig. 2a to include the newly generated data from zygotes, 2-cell and 4-cell embryos. In addition, our principal component analysis (fig. 1e) and clustered heatmaps of enrichment at peaks (fig. 3a) are updated to include the global signal distribution at aggregate H2A.Z peaks. In both cases the new H2A.Z ChIP-seq data from 1- to 4- cell stages provide evidence of gradual progression from an oocyte-like profile towards the enrichment profile seen later in preimplantation embryos. Notably, the H2A.Z profile of zygotes has considerable similarity to that of mature oocyte stages, while the 4-cell profile shows similarity to the 8-cell stage.

We have updated the sentence mentioned by the reviewer to “Overall, H2A.Z was located at genes expressed at ZGA and MGA both during oogenesis and preimplantation embryo development, although the global levels of enrichment were lower at the 2-cell stage”. In addition, we have clarified that we do not claim a direct relationship, by updating lines (197-199) to “H2A.Z may contribute to marking these genes for later transcription *either directly or indirectly* as a placeholder as reported in zebrafish and *Drosophila* embryos”

4. They find an association between oocyte-specific H2A.Z peaks in cluster 3 (Figure 3a) and paternal LADs. They state that oocyte-specific H2A.Z peaks are maternally inherited (line 248)

and thus counteract maternal LAD establishment. However, I don't see the evidence for inheritance as data in zygotes is not shown. Figure 3a shows that cluster 3 is no longer enriched in H2A.Z in 8-cell stage embryos. In any case oocyte-specific peaks could not be maternally-inherited by definition, as they would not be oocyte-specific if they were also present in embryos. Overall, the enrichment of H2A.Z within LADs/iLADs in the embryo is not shown, which limits the interpretation of this data.

Authors comments: Similar to the other comments concerning the gap between MII oocytes and 8-cell embryo H2A.Z ChIP-seq data in the previous submission, then we fully agree that direct data from the actual stages provides stronger and more direct evidence to support our interpretations. The data added for the revised manuscript includes H2A.Z data from zygotes, and it is evident that enrichment in cluster 3 also exists in zygotes (revised Fig. 3a). The point on the definition of oocyte-specific conflicting with maternal inheritance is pertinent, and we have in accordance with this updated the naming of the cluster 1-3 enrichment to 'maternal enrichment'.

Authors actions: The revised manuscript now includes H2A.Z ChIP-seq data from zygotes and the relevant figures have been updated with these, including the point made about cluster 3 (Fig. 3a). We have updated the naming of the cluster 1-3 enrichment to 'maternal enrichment' instead of 'Oocyte specific'. Finally, we have added as requested a new plot showing the overall relationship between H2A.Z peak density and maternal, paternal and shared 2-cell LADs as well as the surrounding regions of the genome (Fig. S3e).

5. An analysis of association between H2A.Z dynamics and H3K4me3/H3K27me3 is shown (Figure 3b), considering the known association between these epigenetic features and the atypical distributions of these marks in early embryos. However, the insight gained from this analysis is limited as the stages during which dramatic changes in distribution of these marks occurs (zygote to 2-cell) is missing.

Authors comments: We agree that critical changes in the distribution of these marks take place at these stages.

Authors actions: We have now added previously published 2-cell H3K4me3 and H3K27me3 data to Figs. 3b, S3c and 4e.

Overall, if the authors wish to make conclusion concerning general patterns of H2A.Z accumulation in oocytes and early embryos and association between H2A.Z incorporation and epigenetic dynamics in early embryos, data of H2A.Z incorporation in early embryos (zygote to 4-cell) should be included to provide a more comprehensive picture of H2A.Z dynamics during these stages of development.

Authors comments: As mentioned above, then we genuinely appreciate the suggestion to include H2A.Z ChIP-seq data from zygotes, 2-cell, and 4-cell embryos. We think that this has strengthened our work.

Authors actions: As mentioned earlier, new zygote, 2-cell, and 4-cell H2A.Z ChIP-seq data has been added during the revision and included in revised figures 1a-c, e, 2a, e, S2a-c, 3a, S3c, 4d, S4a, S6a.

Other major points:

1. Replication timing

When discussing replication timing the authors fail to cite two major papers characterising replication timing during preimplantation development: PMID: 35256805 and 39198647 (lines 253-256). These papers describe the appearance of the replication timing program from the 4-cell stage, in contrast to the 2-cell stage they state. The text should be altered to reflect this. In addition, analysis of replication timing at the 4-cell and 8-cell stage should be conducted in relation to H2A.Z incorporation in oocytes to determine whether the tendency observed in 2-cell stage (Figure 3d) is robust.

Authors comments: We thank the reviewer for pointing out these inadvertent omission of citations, and we agree that including these papers provide a more complete picture of the relevant literature on embryonic replication timing. We also agree that integration of replication timing data from another study is a good way to expand and solidify the observations.

Authors actions: As suggested, we have now added the proposed references to the paper and rephrased the sentence to reflect the three important pieces of work published from Torre-Padilla and Hiratani labs in 2022 and 2024. Here is the revised phrase: “Recent investigations of replication timing in mouse early embryos and 2-cell-like cells revealed a slower DNA replication up until the 8-cell stage and the gradual emergence of a timing program in 2-cell embryos that gets progressively stronger until the 8-cell stage³⁷⁻⁴¹. The timing program correlates with transcription, LADs, genome compartmentalization and inherited histone modifications with consistent differences between the parental genomes^{38,39}.”

In addition, we have performed analyses of the relationship between replication timing in 4-cell and 8-cell embryos and the occurrence of the different clusters of H2A.Z peaks. These analyses have been added in the new Fig. S3g, h. Importantly, these analyses show that both the clusters with embryonic H2A.Z enrichment (clusters 8-10) and those with persistent H2A.Z enrichment (clusters 4-7) are associated with early replication timing in 4-, 8-, and 16-cell embryos (Fig S3g, h) but are generally absent from genomic regions that replicate early in 2-cell embryos, but not 4-cell embryos (Fig S3g). Conversely, peaks from the cluster characterized by late maternal H2A.Z (cluster 3) were more inclined to exist in parts of the genome that were replicated early in 2-cell embryos and less so in 4-cell embryos based on data from Nakatani et al (Nature, 2024) (Fig. S3g).

2. Enhancer analysis.

The authors used a generic list of enhancers from USCS for their analysis. However, many distinct oocyte and embryo-specific enhancers are also proposed to exist (PMID: 38839978). Is H2A.Z enriched in these at the corresponding stages? Is H2A.Z correlated with the presence of H3K27ac at enhancers at the respective stages?

Authors comments: Thank you for pointing our attention to the dataset of putative oocyte enhancers.

Authors actions: We have now added the putative oocyte enhancer analysis to Fig. S4a. We have, however, also kept the general enhancer dataset, as the putative oocyte enhancers are not well-defined and have not been experimentally verified. In addition, it seems that spatial

resolution or precision is lower than the general enhancer set as we observe more enrichment at peripheral locations.

3. Gene expression analysis

How do patterns of H2A.Z incorporation during oogenesis correlate with patterns of RNA expression during oogenesis prior to transcriptional shutdown?

Authors comments: We generally find the link between H2A.Z and transcription to be complex, and we do not see an association between the two that can be explained in simple terms. In our opinion the strongest insights come from the unsupervised clustering of the signal profiles at MTA repeats that generally are expressed and have H2A.Z enrichment (Fig. 5b). This clustering reveals that H2A.Z is elevated at the most transcribed MTAs, but that H2A.Z is juxtaposed to the transcript-rich regions. Moreover, H2A.Z does not occur in silenced regions of the genome (E.g. DNA methylation or H3K27me3 positive MTAs in Fig 5b), so a full understanding of this relationship is complex possibly due to transcriptional activity increasing the eviction rate as we have proposed in a hypothetical model (Fig. 6b).

Authors actions: We have now included RNA data from oocytes to fig. 2b where we correlate H2A.Z and RNA levels at TSSs, as well as an additional figure below showing RNA levels across all H2A.Z peaks in the genome for the reviewer's perusal.

4. TE expression analysis.

The stage-specific association between H2A.Z localization at distinct TE families is intriguing. Many TEs are expressed and act as alternative promoters in these stages of development. A broader analysis of the association between H2A.Z accumulation and TE expression should be conducted. In general, does H2A.Z enrichment correlate or anticorrelate with expression of TEs? Are TE families enriched with H2A.Z at different stages more likely to be expressed or repressed than those without? Does LIT expression correlate with H2A.Z accumulation? Is H2A.Z more or less enriched at insertions more highly expressed and acting as alternative promoters than those that are silent?

Authors comments: We share the fascination of the expression of TE families in oocytes and preimplantation embryos as well as the cooption into transcriptional regulation and protein function. We agree that the multiple copies of almost similar sequences and likely shared overall regulatory mechanisms offers an attractive platform to study interdependences of chromatin states, sequences and expression.

Authors actions: We have expanded Fig. 5b and S6a,b where we compare RNA data to H2A.Z at selected high-copy number TEs that are expressed at the studied stages. While Fig. 5b, S6b includes a panel of histone marks, DNA-methylation, and ATAC-seq in early oocytes, S6a includes RNA-seq from P10 oocytes to 8-cell embryos as well as a range of ATAC-seq datasets from these stages. In short, we find that moderate to highly expressed TEs are associated with nearby H2A.Z, whereas silenced TEs have little or no association. However, the H2A.Z enrichment is often juxtaposed to the expressed areas, and our most simple interpretation is that transcription also counteract H2A.Z possibly through eviction. The relationship between H2A.Z and chromatin accessibility seems more straightforward and positive, although putative nucleosome free regions of course often seem to be depleted of H2A.Z signal, so that H2A.Z enrichment tend to flank the most accessible regions. Finally, we have also compared the level of H2A.Z and H3K4me3 at MTAs overlapping with LITs to those not overlapping with LITs (below). As H2A.Z is only slightly more enriched at LIT MTAs compared to other MTAs, we chose not to include this figure in the paper.

5. Embryo collection

In the methods they describe culture of embryos to E3.5 at which point 8-cell, morula and blastocysts were collected. If this is the case, then the data from 8-cell and morula embryos are not reliable as these embryos are delayed in development and are thus abnormal. 8-cell and morula embryos should be collected at around 70 and 80 h phCG respectively. Please also add the embryo growth protocol to the GEO accessions.

Authors comments: Thank you for making us aware of this error. We are sorry for the incorrect description of collection of some of the embryo stages in the previous version of the manuscript, this was due to misunderstandings between co-authors.

Authors actions: We have changed the methods text to provide the correct descriptions of embryo collection in the revised manuscript. Morulas and blastocysts both were collected at developmental day 3.5 as previously stated, since embryos originated from natural copulation of several mice, and therefore development was not fully synchronized. We have now clarified

this in the text. For 8-cell embryos a new description was added: “8-cell embryos. The same procedure was followed for superovulation and copulation as described above, but with different timing for the hormone injections: PMSG at 3pm and hCG 48h later. At 68h after the hCG injection, the females were euthanized by cervical dislocation and the ovaries with oviducts and upper uterine horns were removed and transferred for processing in a petri dish with M2 medium. One by one, the ovary was removed, the infundibulum was located and slid onto a flushing needle. Embryos were isolated by flushing M2 medium through the oviduct. 8-cell embryos were collected and washed in fresh M2 medium before Tyrode’s acid treatment, further washing in M2, and cross-linking as described below (*picoChIP-seq > Oocyte and embryo cross-linking*).”

We have also added growth protocols to the GEO accessions.

Reviewer #2 (Remarks on code availability):

The authors have made the code available at the above address. However, I am not proficient in coding so I am unable to assess this.

Reviewer #3 (Remarks to the Author):

This study centers on the profiling and analyses of H2A.Z dynamics during oogenesis and preimplantation development. Previous H2A.Z study (PMID 35717671) mainly focused on preimplantation embryos starting from MII eggs to blastocysts. The current study fills the knowledge gap by evaluating H2A.Z dynamics in early growing oocytes including postnatal day 7 (P7), P10, P12, and fully grown oocytes. The study made a few interesting observations including 1) dramatic increase of H2A.Z signal starting from P12 oocytes; 2) strong correlation between H2A.Z and CGI transcription start sites; 3) associations between H2A.Z enrichment and replication timing and lamina associated domains; 4) enrichment of repetitive elements in H2A.Z peaks.

Although this study provides some new insights, I found it largely descriptive and correlative with little functional characterizations. Previous study (PMID 35717671) has 1) established the H2A.Z landscape in preimplantation embryos, 2) demonstrated that H2A.Z is critical for lineage commitment, and 3) characterized how chromatin remodelers may regulate H2A.Z deposition in early development. The current study appears to have a moderate conceptual advance to the field unless functional demonstration of critical roles of H2A.Z in oogenesis.

Authors comments: Thank you for the perceptive review of our manuscript. We appreciate the detailed attention given to our work and the comments provided. However, the previous study mentioned by the reviewer (PMID 35717671), was generated using native ChIP where limited enrichment was observable in oocytes and the earliest embryonic states, whereas our study shows the drastic H2A.Z enrichment at these stages. Furthermore, our study finds pronounced, previously unrecognized, H2A.Z enrichment at TSS-distal sites and an intriguing inverse relationship between H2A.Z and H3K4me3 in low CpG environments, such as MTA and MTB retrotransposons. We also identify late maternal enrichment preceding reduced formation of lamina associated domains and early replication timing in the maternal compared to the paternal genome of 2-cell embryos, indicating that maternal H2A.Z may be instructive for the replication timing in the embryo.

The reviewer is likely to be aware of two new studies recently published after submission of our study in *Nat. Struct. Mol. Biol.*, Mei et al 2025 and Xu et al 2025 (PMID: 40514539 and 40514538). These two studies provide comprehensive and elegant functional characterization of H2A.Z during mouse oogenesis using genetic knockout models. While our work does not include such models, we believe that we present a complementary set of biological observations and inferences increasing the robustness and insight. As an example, Xu et al observe H2A.Z correlation with histone acetylation except at regions with DNA methylation, while we report a similar inverse relationship when analyzing the relationships at several repeat types including MTA, MTB, RMER4B, and RMER6A as well as at CGIs (Fig. 5b,d, S6b). Likewise, Mei et al reports that depletion of ncH3K4me3 by Mll2 knockout also causes a reduction of ncH2A.Z in FGOs, and infers that oocyte ncH2A.Z and ncH3K4me3 reinforce each other, while we systematically dissects the relationship at multiple sets of loci and relates this genome-wide to CpG density showing that P12 H2A.Z and P12 H3K4me3 mainly exist together at high CpG loci (Fig 5a). However, our clustering and analyses of the mutual relationships at high copy number repetitive elements do with high reliability demonstrate that the two marks exist juxtaposed each other, and not at the very same location of the repeat where they rather have inverse relationships (Fig 5b). Altogether this illustrates how KO data taken together with our findings can be used to infer that the causal relationships between H2A.Z and H3K4me3 in growing oocytes are multilayered and more complex than can be observed from KO data alone. Finally, our work provides a more comprehensive set of stages during oocyte maturation and embryo development, including stages that are biologically critical, such as maturation of GVs from NSN to SN and MII. This provides a better insight into the dynamic changes, and in the revised manuscript we have improved this comparative strength further.

Authors actions: For the revised manuscript, we have collected 200 zygotes, 208 cells from 2-cell embryos and 172 cells from 4-cell embryos (see table 1), and generated H2AZ ChIP-seq data from these stages. Our work thereby presents the overall changes in the genome-wide H2A.Z localization with a considerably finer resolution in time and developmental stages. We have also presented a quite comprehensive and strengthened integration and interpretation of the functional context of H2A.Z enrichment at non-TSS loci and relate this to replication timing in the embryo. In our opinion these strengths complement the two studies and the work published in 2022 (PMID 35717671) quite well to provide a more complete overview of H2A.Z in oocyte and embryo development. In the discussion of our revised manuscript, we briefly point the readers' attention to these new pieces of work and address the respective strengths of the works.

Following are some specific comments for authors to consider addressing to further improve the story.

1) Although the bioinformatic analyses describing potential interactions/associations between H2A.Z, CGI, RNA, H3K4me3, H3K27me3, LAD, replication timing, and retrotransposons are interesting, functional characterization is essential to establish causal relationships.

Authors comments: We appreciate the constructive suggestions on how to improve the story, and have now conducted new analyses as well as expanded the scope of a large set of the analyses in the previous submission, to accommodate these suggestions.

Authors actions: We have expanded the functional characterization of the roles of H2A.Z during mouse oocyte and embryo development through new analyses of chromatin accessibility by analyzing ATAC-seq signal in Figures 3b, S3c, S4a, 5b, 5c, 5d, S6a, S6b. Furthermore, we have added analyses of Replication timing in 2, 4, 8, and 16 cell embryos in Figures S3h, S3i, and LAD densities in Figure S3e. We have also added motif densities in Figure S5c, transcript densities from a wider range of developmental stages in Figures 2b, S6a, and S6b, and expanded the histone marks analyzed with H3K27Ac and included H3K4me3 and H3K27me3 data from additional stages in Figures 3b, S3c, and S4a. Additionally we have analyzed H2A.Z profiles at new sets of putative oocyte enhancers in Figure S4a. Figures 1a, 1b, 1c, 1g, 2a, 2e, S2a, S2b, S2c, 3a, S3c, 4d, S4a and S6a have all been updated and expanded to include the new H2A.Z ChIP-seq data from zygotes, 2-cell, and 4-cell embryos and S3a to include new data of replicates from another mouse strain. Finally, we have included confocal imaging and immunofluorescence of H2A.Z as requested below. The result and discussion sections have similarly been updated to include all the new analysis, further strengthening our original conclusions.

2) Immunostaining experiments are recommended to further verify the global increase of H2A.Z from P7/P10 to P12 oocytes.

Authors comments: We agree that it is important to provide independent lines of evidence wherever possible, and immunofluorescence of P7, P10, and P12 oocytes is a good strategy to complement our ChIP-seq data.

Authors actions: We have now, as proposed, collected P7, P10, and P12 oocytes and performed immunofluorescence, confocal microscopy, and quantitation of the fluorescence intensities. The revised manuscript now contains H2A.Z immunofluorescence and confocal microscopy (Fig. 1d) as well as graphs of H2A.Z fluorescence at these stages (Fig. 1e). These new data generally reflect the increase in H2A.Z in P7 to P12 oocytes observed from the ChIP-seq data.

3) How about H2A.Z expression levels in oogenesis? Whether H2A.Z expression changes may explain the H2A.Z dynamics revealed by low-input ChIP-seq?

Authors comments: This is an interesting question that we have also asked during the preparation of the manuscript. To investigate the relationship between H2A.Z expression and the dynamics of its incorporation to the chromatin, we have plotted expression of H2A.Z genes (H2afv and H2afz) in mouse oocytes and preimplantation embryos from publicly available RNA-seq data (PMID: 27626382). According to this plot (see below), H2A.Z gene expressions follow the global transcription in oocytes and early embryos. During MII stage, where the oocytes are transcriptionally silent, H2A.Z RNA level is reduced, followed by a significant increase at the 2-cell stage, when zygotic genome activation takes place. These results are in line with the H2A.Z dynamics revealed by our ChIP-Seq experiments. Since H2A.Z expression follows global transcription, we did not include this plot in the initial submission in order to maintain clarity and

focus on the results that we find are the most interesting and novel.

4) The Fig. 2b RNA FPKM max is quite low (i.e., 3), which may exclude many genes that express higher than this cutoff. As a result, this heatmap could be misleading as it does not show clear patterns of maternal and/or zygotic genes.

Authors comments: Thank you for drawing our attention to this misleading interpretation of the figure. This was not intentional and we have changed the visualization to use log scale to better visualize high transcriptional levels.

Authors actions: We have now updated Fig. 2b with RNA signal shown as log₁₀ FPKM values.

5) The comparison of LAD vs. H2A.Z in Fig3 is confusing. The LAD data were from 2-cell stage whereas none of the clusters in Fig. 3a-b include 2-cell stage.

Authors comments: We understand the reviewer's point of view and as mentioned above, we have now included H2A.Z ChIP-seq from zygotes, 2-cell, and 4-cell embryos. Importantly the relatively high H2A.Z enrichment seen in cluster 3 is preserved from MII oocytes to zygotes. This demonstrates that the H2A.Z enrichment profile is inherited within the maternal genome. We don't want to push conclusions on the new data of the 2-cell stage as the drop in signal intensity is global, and in our experience this can potentially have multiple causes, including biologically driven loss of the target, enrichment variation and a global increase in signal at other loci which due to the compositional nature of the library data leads to a reduced read

count at the studied loci (PMID: 29608657). As the reviewer mentions above, then H2A.Z gene expression changes may explain the H2A.Z dynamics, and the strong transcriptional induction of H2A.Z genes (H2AFV and H2AFZ) together with a higher basal incorporation of H2A.Z throughout the genome could lead to such an effect.

Authors actions: We have now included H2A.Z ChIP-seq from zygotes, 2-cell, and 4-cell embryos.

Minor comments:

1) *Fig. s1c is not clear to readers. What's the interpretation? Whether higher Z-score is better?*

Authors comments: We apologize for the lack of clarity on the purpose of this figure. The Z-score transformation was included to demonstrate the obtained enrichment compared to background, so higher is generally better.

Authors actions: We have clarified this in the text, so it now includes “in terms of FPKM normalized signal and Z-transformed (relative to the entire genome).” (Lines 130-131)

2) *The number of biological replicates is unclear. Table1 suggests that only 1 replicate was generated for each stage.*

Authors comments: Given the scarcity of the material and the time, resources and animal sacrifices this type of data generation requires, it was ethical to balance sacrifices with the gained strength. We did generate two biological replicates of P10 oocytes for the initial submission, and for the revised manuscript, we have now included two additional replicates. For these we used a different mouse strain to demonstrate to ourselves that the H2A.Z enrichment profiles were not dependent on mouse strains. We would also like to note the high degree of correlation between our blastocyst H2A.Z data and previously published blastocyst H2A.Z data (Fig. S1d,e). Finally, the consistency of our data is also demonstrated by the high levels of similarity both in the PCA and unsupervised clustering between our data from several related developmental stages, including P7, P10, and P12 as well as NSN, SN, and MII (Fig. 1g, 3a).

Authors actions: As mentioned above, we have included two additional replicates of H2A.Z ChIP-seq from P10 oocytes to Fig S3a.

POINT-BY-POINT RESPONSE TO REVIEWER COMMENTS

Color codes:

Reviewer comment

Authors' comments

Authors' actions

Reviewer #1 (Remarks to the Author):

The authors have addressed my comments and overall, the paper has significantly improved. However, I have one comment about using ATAC-seq for nucleosome positioning. ATAC-seq has many limitations and it provides rather short fragments (much less than 147 base pairs) and usually ATAC-seq peaks are associated with nucleosome-free regions. So the authors should be very careful in interpreting their ATAC-seq results.

Some wording is still vague. Just one of many examples: "H2A.Z peaks are correlated to H3K4me3" – correlation coefficients can be calculated between the numerical values, not between peaks and modifications. "H2A.Z is strongly 451 correlated to paternal LADs"... "H2A.Z 430 differs markedly from previously assessed epigenetic marks" – H2A.Z is not a modification mark.

Authors' comments

We appreciate the reviewer's positive feedback and are grateful that the improvements are considered significant and largely addressing earlier comments. We agree that the language in the listed examples should be more precise and have gone through the manuscript to identify similar occurrences.

Authors' actions

We have done the following clarifications to narrow the scope of our ATAC-seq interpretations: We added "*chromatin accessibility*" when we describe ATAC-seq the first time (Line 253) and added the second sentence in this passage on lines 416-420: "*Furthermore, H2A.Z and H3K4me3 had minimal colocalization, especially at MTA RTs (Fig. 5b, S6b), where H2A.Z generally occurred more strongly next to the start sites of MTA RTs with high accessibility. This enrichment in ATAC-seq signal combined with the centrally localized absence of H2A.Z signal directly at the MTA start sites might reflect nucleosome-free regions.*"

We have revised the manuscript with more precise wording. Specifically, we have updated the wording in the following sentences:

"Late maternal enrichment was inherited by the zygote, and it preceded reduced formation of lamina associated domains and early replication timing in the maternal genome of 2-cell embryos " (Abstract, Lines 35-37)

"(...)and an intriguing inverse relationship exists between H2A.Z and H3K4me3 signals in low CpG environments,(...)" (Abstract, Lines 40-41)

"While H2A.Z signal intensity at H2A.Z peaks was positively correlated to H3K4me3 levels in P12 oocytes and 8-cell embryos at TSSs (Spearman's $r = 0.83$)" (Line 272-273)

“Thus, the profile of the histone variant H2A.Z differs markedly from previously assessed epigenetic marks at these developmental stages” (Lines 442-444)

“Intriguingly, the maternal H2A.Z signal was strongly associated with absence of LADs and early replication timing in the maternal genome, indicating that maternal H2A.Z incorporation or related features may be instructive for the replication timing in the embryo.” (Lines 463-366)

“Fig. 5 | H2A.Z signal is not correlated to H3K4me3 levels and CpG density at many loci” (Line 1030)

“Fig. S4 | H2A.Z signal intensity is intricately associated to H3K4me3 levels, LADs, and replication timing” (Lines 1082-1083)

“Fig. S6 | H2A.Z signal is correlated to CpG density and found at specific RTs” (Line 1124)

Reviewer #1 (Remarks on code availability):

I did not run the code. I noticed that the README file is almost empty and contains very little information.

Authors' comments

Thank you for drawing our attention to this. We agree that it would benefit from additional information.

Authors' actions

We have now added more explanatory information to the README file, including a link to the manuscript, installation guidance, and a list of dependencies to improve reproducibility.

Reviewer #2 (Remarks to the Author):

I appreciate the authors efforts in adding the new data for zygote to 4-cell stage embryos throughout the manuscript. With this and the additional changes the majority of my comments have been addressed. It appears indeed that the addition of these stages does not change the overall clustering in Figure 3a or the conclusions made regarding persistent, maternal and embryonic clusters. However, in my view it cannot be ruled out that the lack of changes to the overall dynamics is not due to the low library complexity and the inability to call peaks, as they mention. This consideration should be added to the manuscript, for example in the limitations of the study.

Authors' comments

We are once again grateful for the insightful comments made by the reviewer and appreciate that our revision addressed most of the comments raised.

We also appreciate that the reviewer agrees on our interpretations of the clustering, and we acknowledge that improved data depth could affect clustering. Therefore, it is appropriate to mention this as a limitation.

Authors' actions

We have added the following information about the low library complexity to the limitations of the study: *“The library complexity of the zygote to 4-cell stages is lower than that of other developmental stages in this study. This library complexity does not materially affect conclusions about enrichment profiles made at hundreds or thousands of loci, but reduces the signal strength at the single locus level, peak-calling strength, and add stochastic variation to PCA analyses and clustering. Samples with higher library complexities from these stages may therefore improve the understanding of the overall dynamics.”*

Indeed, Figure 3a excludes the data from zygote to 4-cell for the peak clustering (stated in lines 224-225; this should be made clear also in the Figure legend) and thus potential dynamics of these stages may be missed. For example, in 'Fig 3 with new clustering' in the file for the reviewer, cluster 8 now has signal appearing at the 2-cell stage for example. This is apparent with the limited number of called peaks from the new data and thus if more peaks could be called the pattern could change more dramatically.

Authors' comments

As mentioned above, we agree that this should be stated and have updated the “Study limitation” section and legend accordingly.

With regards to the new cluster 8 in the updated peak-set and reclustered figure for the reviewers in the previous point by point response (pasted below for convenience), it is our experience that stochastic variation in e.g. FPKM normalized data often leads to this type of clusters that are seen in a single condition, so we are not certain whether this represents a true positive 4-cell specific enrichment or a aggregated and scaled stochastic variation. Therefore, our conservative judgment was to avoid making conclusions on this.

Fig. 3
a

As Fig. 3 with new clustering
a

Authors' actions

We have added information about zygote-4-cell data being excluded from the peak clustering to the figure 3 legend. (Lines 996-997)

Regarding the replication timing, the reference Nakatani et al (ref 40) should also be cited here (lines 282-284): 'The timing program correlates with transcription, LADs, genome compartmentalization and inherited histone modifications with consistent differences between the parental genomes'.

Authors' actions

We have added the suggested reference at the mentioned section of the manuscript. (Line 295)

Reviewer #3 (Remarks to the Author):

In this revised manuscript, the authors generated new H2A.Z ChIP-seq data for zygotes, 2-cell, and 4-cell embryos, and integrated these data and analyses throughout the manuscript. This addition improves the quality of the study. However, I still have the following major concerns:

Major Comments

1. Data quality in early embryos

I appreciate the authors' effort to generate new data despite the challenges of collecting hundreds of embryos for low-input ChIP. However, the data quality at the zygote to 4-cell stages appears considerably lower than that of other developmental stages, with only ~2 million non-duplicated reads. Consequently, the authors were unable to call peaks from these stages (lines 153-156).

While these data may provide some insights when used in genome-wide analyses, they should be interpreted with caution. For example, the reported global reduction of H2A.Z at TSSs in 2-cell embryos (Fig. 1b and Fig. 2a) may reflect technical limitations rather than biological

differences. I strongly recommend either generating higher-quality datasets for these stages or incorporating already published datasets, as suggested by Reviewer #2.

Authors' comments

We appreciate the comments and agree that the lower library complexity in the zygote to 4-cell embryos is not ideal and we have sought to present this transparently in the manuscript, the previous point-by-point response, and now also in the updated study limitation section and legends. However, while lower library complexity affects peak-calling strength, and adds stochastic variation to PCA analyses and clustering, it is less likely that the mean enrichment at a combined set of hundreds or thousands of loci is affected and it should be proportional regardless of library complexity. So, we believe that complexity does not materially affect conclusions about enrichment profiles.

We agree that adding further cross-validation with other studies is a valuable improvement. Overall, we observe a reassuring level of correlation between comparable developmental stages, despite the lack of signal in MII and the earliest embryonic stages reported by Liu et al. (See figure in actions below) and the differences in underlying ChIP-seq methodology (native vs crosslinked) (New Fig S1d-f, New S2a-c). We also noticed that our method allows for a higher enrichment at oocyte and earliest embryo stages (New Fig S2c).

Authors' actions

We have downloaded and used our own pipeline for processing to enable 1:1 comparison of the newly published oocyte H2A.Z data from Mei et al and Xu et al (PMIDs: 40514539 and 40514538; NSMB 2025). Similarly, we expanded the cross-validation analyses that we had performed earlier to include all the primarily embryonic stages presented in Liu et al (PMID: 35717671).

With these data, we plotted correlations of embryonic H2A.Z at TSSs and throughout the genome, including the requested correlations for Zygotes, 2-cell and 4-cell embryos (New Fig S1d-f). Moreover, we included correlations between our oocyte H2A.Z enrichment profiles and related profiles in the Mei et al and Xu et al datasets (New Fig. S2a,b).

Considering the methodological differences between the native and crosslinked ChIP as well as the oocyte collection protocols, we find a reassuring level of correlation in the data, with correlation coefficients ranging up to 0.78 for Mei et al data (Our P12 oocytes vs Mei et al P15 oocytes) or 0.69 for Xu et al data (our SN vs Xu et al SN oocytes). For comparison the highest level of correlation between related stages for Mei et al and Xu et al data reached 0.8 (FGO vs FGO, see below), and both of these studies were based on native ChIP-seq.

Finally, we performed a clustering based on a combination of our and these published data at TSSs, which also included information about the spatial distribution of the enrichment around the TSSs (New Fig. S2c). This clustering generally reveals a high level of similarity of the H2A.Z distribution near and at the TSSs throughout the tested oocyte and embryo stages, and this also highlighted that our crosslinked ChIP-seq data were able to offer equal or stronger enrichment at P10 to 8-cell stages. We did however notice a larger gap in H2A.Z enrichment directly at the TSSs in the other studies, possibly due to the higher propensity of nucleosomal loss at TSSs in native ChIP (PMIDs: 19633671; 31478357).

The output from these analyses are generally covered in the new text added to the results in lines 141-150.

We have also added the following sentences to the study limitations sections: *“The library complexity of the zygote to 4-cell stages is lower than that of other developmental stages in this study. This library complexity does not materially affect conclusions about enrichment profiles made at hundreds or thousands of loci, but reduces the signal strength at the single locus level, peak-calling strength, and add stochastic variation to PCA analyses and clustering. Samples with higher library complexities from these stages may therefore improve the understanding of the overall dynamics.”*

For your convenience, we include a figure with H2A.Z signal from Liu et al. samples at our peaks below. From this figure, both the correspondence at later embryonic stages and the overall lower H2A.Z signal in the MII to 2-cell stages in the samples from Liu are clearly seen.

2. Lack of biological replicates

With the exception of P10 oocytes, most stages appear to have only one biological replicate.

Although I understand the scarcity of oocytes and early embryos, reproducibility is essential to ensure confidence in the findings. The authors nicely validated their blastocyst data against published datasets (Fig. S1d,e). I suggest applying a similar validation strategy for other stages by comparing with recently published H2A.Z datasets in oocytes and embryos. This would provide cross-validation and strengthen confidence in the conclusions.

Authors' comments

We appreciate the reviewer's point and have sought to cross-validate our data to that of external data on several levels. This is in part explained above and we provide an overview below. We would like to emphasize the similarities in the enrichment profiles, despite the methodological differences. Specifically, the methods used in the studies differs on a range of parameters. Most importantly, our method is the only one that uses crosslinking. While we do not want to promote one method over another, crosslinking may reveal enrichment in genomic regions where the signal may be lost due to 'fragile nucleosomes', such as at some TSSs (New Fig S2c). Besides, these studies differ on the use of fragmentation vs MNase treatment, antibodies, and oocyte stage separation and selection criteria. All in all, we consider this availability of data with methodological diversity a strength for the field that heightens confidence in our conclusions.

Authors' actions

Firstly, we have correlated embryonic signal to that of Liu et al, both at TSSs (New Fig. S1d), and in bins throughout the genome (New Fig. S1e,f). We have then cross-validated our genome-wide signal to that of the two studies published in NSMB during the revision of this manuscript (Mei et al, Xu et al, PMIDs: 40514539 and 40514538; 2025) (New Fig S2a,b). Moreover, we clustered TSS associated enrichment from a selected set of samples from all four studies (New Fig S2c).

Finally, we wanted to demonstrate that the pronounced enrichment of H2A.Z at MTA MaLR retroelements, which we uniquely identify in our work, is also observable in the data from these other studies. Accordingly, we added data from these studies to the clustering of MTAs from New Fig. S7 (see Figure below), and clearly observe similar H2A.Z patterns. Thus, our conclusions can be replicated and validated with data from other studies.

Minor Comment

- A summary table of sequencing metrics (e.g., total reads, mapped reads, deduplicated reads) should be included for all datasets.

Authors' comments

This was originally provided in the reporting summary, but we agree that this should be a more integral part of the work.

Authors' actions

We have made a new table (Table S1, cited in line 133) with the sequencing metrics.

POINT-BY-POINT RESPONSE TO REVIEWER COMMENTS

Color codes:

Reviewer comment

Authors' comments

Authors' actions

Reviewer #1 (Remarks to the Author):

My comments have been addressed.

Reviewer #1 (Remarks on code availability):

the description of codes is still limited. A log list of scripts but it is not clear what they are for.

Authors' comments: We would like to thank the reviewer for the feedback. The limited code description is not intentional, and we have now provided additional information that may be of use.

Authors' actions: We have updated the GitHub page with additional information to improve clarity and reproducibility. Specifically, we have now added description for each script for their specific use, added a usage section on how to clone and navigate to the repository, added instructions on how to install the required R packages inside R through CRAN and Bioconductor and finally polished the formatting of the readme file.

Reviewer #2 (Remarks to the Author):

My outstanding comments have now been addressed and I can recommend publication of this manuscript.

Authors' comments: We are very pleased to hear this and thankful for the time dedicated for this review.

Reviewer #3 (Remarks to the Author):

The authors have conducted extensive comparisons between their datasets and previously published datasets, as I had suggested. These analyses are informative and help guide readers in accurately interpreting results generated using different methodologies. For example, crosslinking appears to cause stronger enrichment of H2A.Z at transcription start sites (TSS) compared with native ChIP conditions (new Fig. S2c). This may also explain why Liu et al. detected relatively weak H2A.Z signals at the MII stage, which used a native ChIP condition.

I appreciate the authors for performing these additional analyses and for providing a comprehensive comparison with prior datasets generated using distinct experimental approaches. I have no further concerns and support publication of this work.

Authors' comments: We are very pleased to hear that the reviewer finds the new additions comprehensive, informative and helpful, and that the reviewer values having complementary experimental approaches. We are grateful for the time dedicated to the earlier feedback.